# *Lo-Hi*: Practical ML Drug Discovery Benchmark

**Simon Steshin**
Independent Researcher
`simon.steshin@gmail.com`

## Abstract

Finding new drugs is getting harder and harder. One of the hopes of drug discovery is to use machine learning models to predict molecular properties. That is why models for molecular property prediction are being developed and tested on benchmarks such as MoleculeNet. However, existing benchmarks are unrealistic and are too different from applying the models in practice. We have created a new practical *Lo-Hi* benchmark consisting of two tasks: Lead Optimization (Lo) and Hit Identification (Hi), corresponding to the real drug discovery process. For the Hi task, we designed a novel molecular splitting algorithm that solves the Balanced Vertex Minimum $k$-Cut problem. We tested state-of-the-art and classic ML models, revealing which works better under practical settings. We analyzed modern benchmarks and showed that they are unrealistic and overoptimistic.

Review: `https://openreview.net/forum?id=H2Yb28qGLV`
Lo-Hi benchmark: `https://github.com/SteshinSS/lohi_neurips2023`
Lo-Hi splitter library: `https://github.com/SteshinSS/lohi_splitter`

## 1 Introduction

Drug discovery is the process of identifying molecules with therapeutic properties [1]. To serve as a drug, a molecule must possess multiple properties simultaneously [2]. It must be stable [3], yet easily eliminated from the body [4], able to reach its target [5, 6, 7], non-toxic [8], cause minimal side effects [9], and therapeutically active on the target [10, 11].

To identify such molecules, researchers develop Molecular Property Prediction (MPP) models. These models are used in a virtual screening, during which the models are used to make predictions for a large number of molecules, after which the molecules with the best estimated property are selected for experimental validation [12, 13, 14, 15, 16].

To enable comparisons between various architectures, the research community relies on standardized benchmarks to evaluate model performance [17, 18, 19, 20, 21, 22, 23]. The prevailing assumption is that models with superior metrics on these benchmarks are more suitable for real-world applications.

We believe this assumption is false. Modern benchmarks test in non-realistic conditions on impractical tasks. In this paper, we introduce two new practical drug discovery ML tasks — Hit Identification and Lead Optimization — that are often encountered in most drug discovery campaigns. We demonstrate that none of the benchmarks assess models for these tasks, which is why we propose seven new practical datasets that better imitate real-life drug discovery scenarios. To prepare Hi datasets, we designed a novel molecular splitter algorithm that solves Balanced Vertex Minimum $k$-Cut problem using Integer Linear Programming with heuristics.

37th Conference on Neural Information Processing Systems (NeurIPS 2023) Track on Datasets and Benchmarks.

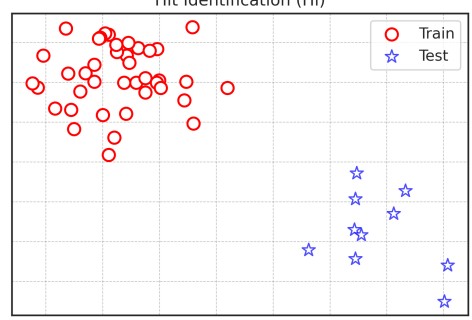

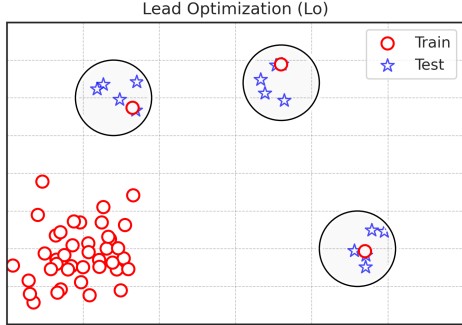

Figure 1: Hit Identification (Hi) task    Figure 2: Lead Optimization (Lo) task

## 2 Practical drug discovery

### 2.1 Hit Identification (Hi)

Drug discovery involves multiple stages. When the potential mechanism of action is known, an early step is Hit Identification. During this phase, chemists search for a "hit" - a molecule with the potential to become a drug [24]. A viable hit must exhibit some level of activity towards the target (e.g., Ki less than 10 $\mu$M) and possess novelty, meaning it is eligible for patent protection.

Clinical trials can incur costs in the hundreds of millions of dollars [25, 26], making it risky to pursue non-patentable molecules. Consequently, companies prioritize patentable molecules from the outset. Novelty is an essential aspect of this process. In practice, medicinal chemists often pre-filter their chemical libraries before the virtual screening, eliminating molecules that exhibit a Tanimoto similarity above a specific threshold to molecules with known activity [10, 11, 16, 27, 28, 29].

### 2.2 Lead Optimization (Lo)

Following Hit Identification, the next stage is Lead Optimization. Once a novel, patentable molecule with activity towards the desired target is identified, its closest analogs are also likely to exhibit activity [30]. In this phase, medicinal chemists often make minor modifications to the hit molecule to enhance its target activity, selectivity, and other properties [31, 32]. Unlike in Hit Identification, novelty usually is not a priority during Lead Optimization. Instead, the objective is to discover molecules that are similar to the original hit but possess improved characteristics.

This is related to the field of goal-directed molecular generation [33], which is occasionally formulated as the problem of searching for molecules with maximal activity within the $\varepsilon$-neighborhood of a known hit [34, 35], within some scaffold [36], or as an optimization process in the latent space [37]. These works use predictive ML models to distinguish between minor chemical modifications [38]. The effectiveness of these models in their ability to distinguish between molecules with small modifications remains unclear.

### 2.3 Model Selection

In this context, Lead Optimization (Lo) and Hit Identification (Hi) represent contrasting tasks. Hi ML models are expected to demonstrate their generalization capabilities by predicting properties of molecules significantly different from the training set. Conversely, Lo ML models should predict properties of minor modifications of molecules with known activity.

For selecting the appropriate ML model or adjusting its hyperparameters, it is crucial to test the model under conditions similar to its intended application. In Hi, models must identify novel molecules markedly different from known active ones, implying the models should be tested on molecules distinctly different from the training set. Conversely, Lo applies ML models to molecules resembling known active ones. The null Structure-Activity Relationship hypothesis assumes that

small modifications will not alter the molecule's properties [30, 39]. Consequently, Lo ML models should be evaluated based on their ability to make predictions that surpass the null hypothesis.

We show that modern benchmarks mix these two scenarios, which is why it remains unclear how effectively models can generalize to truly novel molecules and how proficiently they can distinguish minor modifications, thereby guiding molecular optimization. Furthermore, we untangle these scenarios using a novel benchmark, demonstrating that **different architectures are better suited for different tasks**.

## 3   Novelty

But how to measure novelty? From a commercialization and regulatory perspective, a novel molecule is one that can be patented. Contemporary patents consist of human-written text and incorporate non-trivial substitutions, rendering the assessment of a molecule's patentability a complex task for legal professionals. To date, this process has not been automated. Additionally, during the Hit Identification, not only must the hit be patentable — its neighborhood should be patentable as well to make Lead Optimization possible. When dealing with an extensive chemical library, these challenges make patentability an impractical criterion for determining novelty.

While chemists agree that minor substitutions likely do not alter a molecule's function, a consensus on the distinction between a new molecule and a modified version of an existing one remains elusive. One might hope to find a general similarity threshold that would separate similar molecules with presumably the same activity from distinct molecules. This hope led to the "0.85 myth" [40], which proved to be false due to the significant variability of such thresholds across different targets [41]. About molecular similarity, it was said [42, 41], "Similarity is in the eye of the beholder." Nevertheless, a practical criterion exists.

In study [43] 143 experts from regulatory authorities(FDA, FDA of Taiwan, EMA, PMDA) were shown 100 molecular pairs and asked if the molecules should be regarded as structurally similar. This study is especially notable because it simulates a real-life scenario in which a regulatory committee (EMA's Committee for Medicinal Products for Human Use) decides if a new drug is novel enough to grant it a beneficial status: orphan drug designation. The study shows that when two molecules have Tanimoto similarity $\approx 0.4$ with ECFP4 fingerprints [44], half of the experts regard them as dissimilar, which is sufficient to conclude the novelty of the drug. See Appendix G for our reproduction.

While Tanimoto similarity with ECFP4 fingerprints has its own disadvantages such as size bias [45], its simplicity makes it possible to quickly measure the similarity of molecules, so there are practical [16] and theoretical [46, 47] works that use "Tanimoto similarity < 0.4" as a novelty criterion. Because of the practical evaluation in real-life scenarios and efficiency, we assess novelty using Tanimoto similarity with ECFP4 fingerprints in our benchmark.

## 4   Our contribution

- We suggest two new practical drug discovery ML tasks — Hit Identification and Lead Optimization — that better imitate real-life scenarios;
- We designed a novel molecular splitting algorithm for the Hi task;
- We propose seven new practical datasets;
- We demonstrate that modern ML drug discovery benchmarks simulate impractical scenarios;
- We evaluate modern and classic ML algorithms on our benchmark.

## 5   Modern benchmarks test neither Lo nor Hi

To select a model or its hyperparameters, we must evaluate them under the conditions in which they will be used. We demonstrate that none of the modern benchmarks correspond to realistic conditions, thus raising questions about their suitability for evaluating practical machine learning models. Although it is impossible to examine every existing benchmark due to their sheer number, we have opted to assess a variety of benchmarks using different data, different preprocessing techniques, and originating from different authors. We will initially focus on standard benchmarks, while in

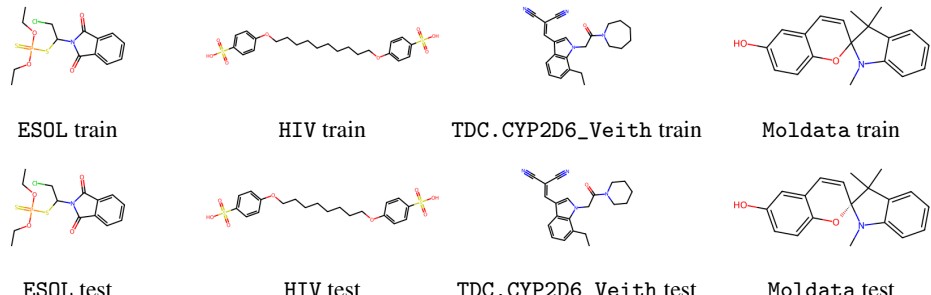

ESOL train      HIV train      TDC.CYP2D6_Veith train      Moldata train

ESOL test      HIV test      TDC.CYP2D6_Veith test      Moldata test

Figure 3: Common benchmarks contain highly similar molecules across different splits.

the section 6, we will explore more exotic and specialized ones. We provide additional analysis in Appendix D.

### 5.1 MoleculeNet: ESOL

MoleculeNet [17] is a widely-used benchmark for ML drug discovery, consisting of 17 datasets, including the ESOL dataset with water solubility data. This dataset is frequently used in studies for model comparisons [48, 49, 50, 51, 52, 53]. While the authors recommend a random train-test split, we discovered that it leads to 76% of the test molecules having a neighbor in the train with Tanimoto similarity > 0.4, making ESOL unsuitable for Hi scenario (see Fig. 3). It is also unclear how well the dataset represents the Lo scenario, as the distinct 24% of the train contributes to the evaluation, and the RMSE metric can be significantly improved over the constant baseline, even without identifying minor chemical modifications.

### 5.2 MoleculeNet: HIV

In benchmarks, train and test sets are typically split using random, scaffold, or occasionally time splits when time stamps are available. It has been frequently observed that random splits can result in highly similar molecules in both train and test sets, leading to overly optimistic and impractical estimations. For example: "Random splitting, common in machine learning, is often not correct for chemical data" [17]. Thus, scaffold splitting [54] is sometimes suggested as an alternative.

Scaffold splitting [54] is a method where each molecule is represented by a graph consisting of ring systems, linkers and side chains. A group of molecules may correspond to a single graph, in which case they are assigned to the same partition.

The MoleculeNet HIV dataset (ubiquitous in evaluation [55, 56, 57, 58, 59]) recommends using scaffold splitting, as it is believed to better reflect the process of discovering new molecules: "As we are more interested in discovering new categories of HIV inhibitors, scaffold splitting [...] is recommended" [17]. Although scaffold splitting makes the train set more distinct from the test set, it is still insufficient for the scenario of finding novel molecules. We discovered that 56% of HIV test molecules have a train neighbor with a Tanimoto similarity > 0.4, indicating the dataset is unsuitable for the Hi scenario (see Fig. 3). Additionally, the dataset is unsuitable for the Lo scenario due to its binary label, rather than a continuous value.

### 5.3 Therapeutic Data Commons: TDC.CYP2D6_Veith

The Therapeutic Data Commons [18] is a novel platform designed for evaluating machine learning models in drug discovery. We examined the largest dataset, TDC.CYP2D6_Veith, within the Single-Instance Learning Tasks category. Although a scaffold split is recommended, the dataset is deemed unsuitable for the Hi scenario, as 78% of test molecules possess a training neighbor exhibiting a Tanimoto similarity greater than 0.4 (see Fig. 3). Additionally, the dataset is unsuitable for the Lo scenario due to its binary label, rather than a continuous value.

### 5.4 MolData

`MolData` [19] is a recent benchmark for MPP based on PubChem data. The authors themselves analyzed the novelty of the molecules within the subset and found their scaffold split to be unrealistic: "[...] more than 44% of the molecules within the `MolData` dataset have at least one other similar molecule to them with a Tanimoto Coefficient of 0.7 or higher. This high percentage of the similarity can denote lack of diversity within this portion of the dataset."

The authors use scaffold split to increase the difference between the train and the test, but we found that at least 88% of the molecules in the test have a similar molecule in the train with a Tanimoto similarity > 0.4, making the dataset unsuitable for the Hi scenario (see Fig. 3). Additionally, the dataset is unsuitable for the Lo scenario due to its binary label, rather than a continuous value.

## 6 Related works

### 6.1 Out-of-distribution MPP

In the Hi scenario, the training and test sets are significantly different to simulate the search for new molecules, making the Hi scenario interpretable as an Out-Of-Distribution (OOD) task. Although OOD benchmarks already exist for machine learning-based drug discovery, they do not align with practical drug discovery applications.

`DrugOOD` [20] is a drug discovery benchmark dedicated to Out-Of-Distribution prediction. The authors accurately observe that "In the field of AI-aided drug discovery, the problem of distribution shift [...] is ubiquitous", and therefore suggest assay-based, scaffold-based, or molecular size-based partitions to create a distribution shift. However, their benchmark does not correspond to practical drug discovery, as it involves predicting average activity for all ChEMBL targets simultaneously in ligand-based drug discovery datasets. This approach is impractical because, in reality, researchers are not interested in average activity across ChEMBL targets, but rather in activity on a specific target.

`GOOD` [21] is a benchmark explicitly designed to separate covariate and concept shifts. We examined the `GOOD-HIV` dataset, as it is a MPP benchmark containing a significant amount of diverse data (see the #Circles in Appendix A). In all Out-Of-Distribution test partitions, more than 40% of molecules have a neighbor in the training set at a Tanimoto similarity more than 0.4, making the dataset unsuitable for the Hi scenario.

During the NeurIPS review period, Valence Labs and Laval University jointly published work — independent from ours — investigating Molecular-Out-Of-Distribution generalization [60]. Their research is akin to our Hi-scenario, but the authors studied different data splits, employed different models, and investigated uncertainty calibration.

### 6.2 Activity cliff

Activity cliff is a pair of structurally similar compounds that are active against the same bio-target but significantly different in binding potency [61, 22].

Recently, several remarkable studies have emerged, seeking to establish a standard benchmark for Activity Cliff Prediction [22, 23]. These works have inspired us to create our own benchmark. While Activity Cliff Prediction may be advantageous in the Lo scenario, it is inadequate for guiding molecule optimization. In the Lo scenario, it is essential not only to identify extreme activity cliffs with a 10x difference in activity but also to predict minor activity fluctuations. We propose alternative splits and metrics in the Lo scenario.

## 7 Results

### 7.1 Hi-splitter

To simulate the Hi scenario, it is necessary to divide the dataset into training and testing subsets such that any pair of molecules from different partitions has a ECFP4 Tanimoto similarity of less than $0.4$. Despite the fact that the scaffold splitter produces a more diverse division than random splitter, we observe that it is inadequate for an effective Hi scenario.

A greedy approach is to implement the conventional scaffold split and discard from the test set any molecules excessively similar to those in the training set. The issue with this method, however, is that data points are costly, and it is desirable to minimize the number of discarded molecules. Consequently, we have developed a novel algorithm for strict dataset splitting, which discards fewer molecules than the greedy algorithm.

Let us consider molecules $X = \{x_1, ..., x_n\}$. We construct a neighborhood graph $G = (V, E)$, where each molecule $x_i$ corresponds to a vertex $v_i \in V$. Two vertices are connected by an edge if and only if the associated molecules have a similarity to each other greater than threshold $t$: $e_{vu} \in E \Leftrightarrow T(x_v, x_u) > t$, where $T$ is Tanimoto Similarity with ECFP4 fingerprints. In such a graph, connected components can be assigned to the training or testing sets independently. However, in practice, 95% of the molecules belong to a single connectivity component. Our goal is to remove the minimum number of vertices such that the giant component breaks up into multiple components with size constraints, thereby enabling us to distribute them between the training and testing sets.

This problem is known as the Balanced Vertex Minimum $k$-Cut and has been extensively researched in literature [62, 63, 64]. A review of similar problems can be found elsewhere [65, 66]. Similar to [62] we formulate the Integer Linear Programming formulation.

Let $K$ denote the set of integers $\{1, 2, \ldots k\}$. Let $G = (V, E)$ represent a simple connected graph that we are going to split:

$$V = \cup_{i=1}^{k} V_i \cup V_0 \qquad \forall i \neq j \quad V_i \cap V_j = \emptyset$$

For all vertices $v \in V$ and for all integers $i \in K$, let us associate a binary indicator $y_v^i$ such that:

$$y_v^i = \begin{cases} 1, & \text{if } v \in V_i \\ 0, & \text{otherwise} \end{cases}$$

Note that $V_0$ denotes the set of removed vertices, so if $\sum_{i \in K} y_v^i = 0$ then the vertex $v$ is in $V_0$.

Let $b_i \in \mathbb{N}$ be a lower bound on the cardinality $|V_i|$ of partition $V_i$. We need it to get partitions with size constraints, e.g. 80% in train and 20% in test. Let $w_v$ be the weights of the nodes. In simple formulation we take $w_v = 1$. We formulate the Balanced Vertex Minimum $k$-Cut as follows:

$$\max \sum_{i \in K} \sum_{v \in V} w_v y_v^i \tag{1}$$

$$\sum_{i \in K} y_v^i \leq 1 \qquad \forall v \in V \tag{2}$$

$$y_u^i + y_v^j \leq 1 \qquad \forall i \neq j \in K, \forall e_{uv} \in E \tag{3}$$

$$\sum_{v \in V} y_v^i \geq b_i \qquad \forall i \in K \tag{4}$$

$$y_v^i \in \{0, 1\} \qquad \forall i \in K, \forall v \in V \tag{5}$$

Equation (1) minimizes the weight of removed molecules $V_0$. Equation (2) states that each vertex $v$ should be in one partition maximum. Equation (3) ensures there is no connectivity between different partitions, so for each edge $e_{uv} \in E$ if $u$ is in $V_i$ partition, then the other vertex $v$ is either in $V_i$ as well, or in $V_0$, meaning it was removed. Equation (4) puts constraints on the size of the partitions $V_i$.

While this formulation was effective on small-scale graphs (around 100 vertices), we found it too slow for a real DRD2 activity dataset with 6k molecules. To our knowledge, existing literature [62, 65, 67, 68] typically involves small graphs from the standard benchmarks with a maximum node count of several hundred. To expedite computation, we implemented a graph coarsening approach. The basic idea is outlined here, while the formal algorithm is detailed in Appendix E.

We initially performed Butina clustering [69] on molecules and created a coarse graph wherein the vertices correspond not to individual molecules but to clusters of molecules. In the coarsened graph,

each vertex is assigned a weight $w_v$ equal to the number of molecules in the respective cluster. This approach accelerated computations and enabled us to partition the HIV dataset consisting of 40k molecules, removing fewer vertices than would be the case with the greedy approach (See Table 1).

Table 1: Number of removed molecules for 0.9:0.1 split

| Method | DRD2 | HIV |
|---|---|---|
| Greedy | 1066 (17.0%) | 5851 (14.2%) |
| Hi-Splitter | 97 (1.5%) | 1598 (3.8%) |

## 7.2 *Lo-Hi* benchmark

We conducted an analysis of numerous drug discovery benchmarks, yet none seemed to align with the actual drug discovery process. Consequently, we prepared seven datasets and are now making them available to the community.

We selected datasets that represent realistic drug discovery problems, contain substantial amounts of qualitative data, and cover a diverse chemical space. Diversity was assessed using the recently introduced #Circles metric [70]. As the source code was unavailable at the time of writing, we employed our own implementation of the greedy algorithm outlined in [70]: Appendix H. We provide additional statistics for the original datasets in the Appendix A.

We propose three distinct train-test splits for each dataset. We advise adjusting hyperparameters solely on the first split, applying the same hyperparameters to train and assess models on splits #2 and #3, and comparing models by averaging metrics across the splits. Datasets are released under the MIT license.

### 7.2.1 Hi

In the Hit Identification scenario, we aim to predict binary labels for new molecules that significantly differ from those in the training set. We prepared four datasets which we divided into training and testing sets, ensuring that the Tanimoto similarity between any molecule in the test set and those in the training set is less than 0.4. See Fig. 1. Additionally, we show that such a split predicts experimental outcomes better than the scaffold split (Appendix F).

DRD2-Hi involves predicting dopamine receptor inhibition, a GPCR target of therapeutic importance in schizophrenia [71, 72] and Parkinson's disease [73, 74]. To create this dataset, we obtained Ki data for DRD2 from ChEMBL30 [75], cleaned it (see Appendix A), and binarized it so that molecules with Ki < 10μM are considered active.

HIV-Hi is an HIV dataset from the Drug Therapeutics Program AIDS Antiviral Screen that measures the inhibition of HIV replication. We obtained the prepared dataset from MoleculeNet [17].

DRD2-Hi and HIV-Hi are large. However, real-life data often comes in limited quantities. To simulate this crucial scenario, we prepared a smaller challenging dataset, KDR-Hi. This dataset is based on the ChEMBL30 IC50 data associated with vascular endothelial growth factor receptor 2, a kinase target for cancer treatment [76]. Its creation process was similar to that of DRD2-Hi, but we restricted the training folds to just 500 molecules.

The Sol-Hi dataset draws from a public solubility dataset at Biogen [77]. We binarized this data such that molecules with a solubility of less than 10 μg/mL were assigned a positive label.

Each dataset was divided into distinct training and testing sets. We used the Hi-splitter with $k = 3$ to obtain three highly dissimilar subsets: $\{F_1, F_2, F_3\}$. These subsets were then combined to create three distinct folds, each with a unique test set. For instance, for the first fold, the training set was $train_1 = \{F_1, F_2\}$ and the test set was $test_1 = \{F_3\}$. This methodology enabled us to assess the variability in quality resulting from using different data with the same models.[1]

**Hi Metric** For our benchmark, we have selected the PR AUC. As a simple binary classification metric without parameters, it is implemented in most libraries and normalized to a range of [0, 1].

---

[1]The DRD2-Hi preparation code can be found at `notebooks/data/03_split_drd2_hi.ipynb`.

The PR AUC favors early recognition models [78] and does not appeal to wrong intuition among readers in an unbalanced setting.

### 7.2.2 Lo

In the Lead Optimization scenario, we aim to predict the activity of molecules that are highly similar to those in the training set. As similar molecules tend to exhibit similar activity, our focus is not on predicting binary labels, but rather on ranking, which indicates whether a modification increases the activity or not.

To simulate the Lead Optimization scenario, we isolated clusters of molecules with intracluster similarity $\geq 0.4$ and consisting of $\geq 5$ molecules. We included them in the test dataset. In practical Lead Optimization, the activity of a given hit is already known; thus, for each cluster, we retained exactly one molecule with a similarity $\geq 0.4$ to that cluster in the training set. See Fig. 2. We provide additional analysis in Appendix A.

In order to confirm the validity of the Lo benchmark, we ensured that the intracluster variation in activity exceeds the experimental noise (See Appendix B). This step is critical, as if the variance within a cluster of similar molecules is not significantly greater than the experimental noise, it would not make sense to test models on such molecules — under these conditions, even an ideal model would struggle to make accurate predictions. The formal pseudocode can be found in Appendix C.

We assembled three datasets: `DRD2-Lo`, `KCNH2-Lo` and `KDR-Lo`. Both `DRD2-Lo` and `KDR-Lo` are based on the same data as their respective `Hi` counterparts but feature a different split between training and testing sets. KCNH2 is an ion channel that regulates heartbeat, and its inhibition can cause dangerous side effects [79]. Consequently, bioassays for KCNH2 are used as a screening method for cardiotoxicity. We extracted IC50 data from ChEMBL30, cleaned it, and divided it into training and testing sets.

Each dataset was split three times using different random seeds, resulting in three folds. This method enables us to assess the variability in quality when using different data with the same models.[2]

**Lo Metric**     Our goal is to determine whether the models can make better predictions than assuming "the modified molecule active in the same manner as the original hit." We chose Spearman's correlation coefficient as our metric, calculated within each cluster and averaged across clusters. This metric does not rely on intracluster variation, depends solely on the ranking of molecules within the cluster, and is normalized, between minus one (ideally wrong), zero (random) and one (ideal), rendering it easily interpretable.

### 7.3 Evaluation

We evaluated both traditional and state-of-the-art ML models on our benchmark. We meticulously executed the hyperparameter search, following the procedure outlined in Appendix H. Results are presented in Table 2. The best scores for fingerprint and graph models are highlighted in bold. It should be noted that the mean and standard deviation were calculated not for random seeds, but within different folds.

We found that the best models varied for the Hi and Lo tasks. The most effective model for the Hi task was the Chemprop graph neural network [80], aligning with its real-world success [16, 28, 29]. The second-best were gradient boosting and KNN for the small `KDR-Hi` dataset.

Conversely, for the Lo task, the SVM model was found to be the most proficient, which corroborates previous works on activity cliffs [23]. Only for the small `KDR-Lo` did Chemprop outperform SVM, but even then, the performance was not satisfactory. We further provide a per-cluster Spearman distribution for SVM in Appendix I. The limited inability of Chemprop to distinguish minor modifications may be due to the limited expressivity of graph neural networks. However, this was surprising to us, considering the limited expressivity of binary fingerprints as well.

---

[2]The DRD2-Lo preparation code can be found at `notebooks/data/04_split_drd2_lo.ipynb`.

Table 2: *Lo-Hi* results. PR AUC for `Hi` and mean Spearman for `Lo`.

| Model | DRD2-Hi | HIV-Hi | KDR-Hi | Sol-Hi | DRD2-Lo | KCNH2-Lo | KDR-Lo |
|---|---|---|---|---|---|---|---|
| Dummy baseline | 0.677±0.061 | 0.04±0.014 | 0.609±0.081 | 0.215±0.008 | 0.000±0.0 | 0.000±0.0 | 0.000±0.0 |
| KNN (Tanimoto distance, ECFP4) | 0.706±0.047 | 0.067±0.029 | **0.646±0.048** | 0.426±0.022 | 0.195±0.053 | 0.164±0.014 | 0.130±0.034 |
| KNN (Tanimoto distance, MACCS) | 0.702±0.042 | 0.072±0.036 | 0.610±0.072 | 0.422±0.009 | 0.211±0.041 | 0.036±0.022 | 0.071±0.02 |
| Gradient Boostring (ECFP4) | 0.736±0.05 | **0.08±0.038** | 0.607±0.067 | 0.429±0.006 | 0.145±0.052 | 0.37±0.003 | 0.076±0.036 |
| Gradient Boostring (MACCS) | **0.751±0.063** | 0.058±0.03 | 0.603±0.074 | **0.502±0.045** | 0.197±0.043 | 0.216±0.032 | 0.100±0.026 |
| SVM (ECFP4) | 0.677±0.061 | 0.04±0.014 | 0.611±0.081 | 0.298±0.047 | **0.311±0.015** | **0.472±0.014** | **0.158±0.051** |
| SVM (MACCS) | 0.713±0.05 | 0.042±0.015 | 0.605±0.082 | 0.308±0.021 | 0.219±0.02 | 0.133±0.024 | 0.074±0.034 |
| MLP (ECFP4) | 0.717±0.063 | 0.049±0.019 | 0.626±0.047 | 0.403±0.017 | 0.094±0.059 | 0.146±0.040 | 0.085±0.030 |
| MLP (MACCS) | 0.696±0.048 | 0.052±0.018 | 0.613±0.077 | 0.462±0.048 | 0.026±0.083 | 0.174±0.031 | 0.065±0.027 |
| Chemprop [80] | **0.782±0.062** | **0.148±0.114** | **0.676±0.026** | **0.618±0.03** | **0.298±0.035** | **0.375±0.067** | **0.161±0.024** |
| Graphormer [81, 82] | 0.729±0.039 | 0.096±0.070 | - | - | - | - | - |

# 8 Conclusion

In virtual screening, practitioners aim to discover novel molecules and filter their chemical libraries using Tanimoto similarity. Despite this common practice, there are no benchmarks that simulate this particular scenario. In molecular optimization, researchers employ predictive models to guide the optimization process in a step-by-step manner. However, it remains unclear whether these models possess the capacity to distinguish minor modifications.

We identified several limitations in current drug discovery benchmarks and proposed a more realistic and practical alternative in the form of the *Lo-Hi* benchmark. By introducing two tasks, Lead Optimization (Lo) and Hit Identification (Hi), which closely resemble real drug discovery scenarios, we created an environment to evaluate machine learning models under more representative conditions. We emphasize the importance of testing models under conditions similar to their intended application.

Furthermore, the paper critically assesses existing benchmarks and related works, highlighting their inadequacies and the need for better evaluation methods. To address these issues, we suggest alternative datasets. To build them, we designed a novel molecular splitter algorithm for the Hi task.

Different models proved to be better suited to different tasks. Our evaluation of both classical and modern ML models revealed Chemprop as the state-of-the-art for the Hi task, and SVM with ECFP4 fingerprints as the state-of-the-art for the Lo task.

The paper's key contributions include the introduction of the *Lo-Hi* benchmark, a comprehensive analysis of the limitations of modern ML drug discovery benchmarks, a novel molecular splitter algorithm and the evaluation of modern and classic ML algorithms on the proposed benchmark. This work sets the stage for a more accurate and reliable evaluation of machine learning models in the field of drug discovery, ultimately leading to better decision-making and improved outcomes in the search for new therapeutic compounds.

# 9 Limitations

We have conducted hyperparameter tuning, although performing it thoroughly poses a challenge [83, 84, 85]. Convincing evidence supporting a particular architecture could be garnered from an open online contest with prizes, accompanied by an undisclosed test dataset. We faced numerous technical difficulties in executing and modifying Graphormer (see Appendix H.7). As such, we cannot definitively determine if Graphormer's failure is a consequence of its architecture or the result of improper dependency pinning by the authors.

Our findings indicate that the Integer Linear Programming solution proves to be significantly more effective than the greedy approach. However, we have not explored different formulations in our study. Therefore, it is possible that more efficient methods for splitting molecular datasets could exist.

We endeavored to encompass a diverse range of ligand-based drug discovery problems (GPCR inhibition, kinase inhibition, cardiotoxicity, solubility) in our benchmark. However, it is infeasible to capture every potential molecular property prediction task. We advise practitioners to use benchmark results to shortlist models, but also to test them against specific objectives.

## 10    Future work

Our focus was on medium and large datasets, yet many small datasets contain fewer than 100 data-points. It would be beneficial to have smaller datasets similar to [86] but tailored for Hi generalization.

While our emphasis was on ligand-based drug discovery, where the goal is to predict a molecule's property, there is also structure-based drug discovery. This approach not only involves predicting molecular properties but also incorporates protein information. Hence, it would be advantageous to have structure-based drug discovery datasets that are divided not just by protein (or pockets [87]) similarity but also with Hi generalization across ligands.

A major ongoing challenge in molecular generative models is ensuring synthesizability, meaning that generated molecules can be made in the real world. Hi splits can help test the generalizability of synthesizability models. But, it's important to remember that Lo/Hi splits assume similar molecules have similar properties. While this holds for physico-chemical attributes, this premise remains to be validated in the context of feasibility measures.

## 11    Potential Harmful Consequences

One of the primary concerns arises from the ability to design molecules that are unfamiliar to medical chemists and experts in the field. While the novelty of these compounds can be advantageous for pushing the boundaries of current scientific knowledge, it also raises the potential risk of misuse, especially if malicious actors were to use the system to generate harmful or toxic compounds for hostile purposes. Given their unfamiliar nature, these molecules might not immediately raise flags upon review by experts or during synthesis orders at chemical laboratories. This could open the door for the creation and dissemination of harmful compounds.

## 12    Acknowledgements

This work was funded by Gleb Pobegailo.

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

# A  *Lo-Hi* benchmark

*Lo-Hi* is a practical ML drug discovery benchmark[3], comprising two tasks: Hit Identification (Hi) and Lead Optimization (Lo). Hi corresponds to a binary classification problem, wherein the goal is to identify novel hits that differ significantly from the training dataset [10, 11, 16, 27, 28]. This is why there are no molecules in the test set with ECFP4 Tanimoto similarity exceeding $0.4$ to the training set. Models are compared using the PR AUC metric.

Lo is a ranking problem that pertains to optimizing molecules or guiding molecular generative models. The test set consists of clusters of similar molecules that are largely dissimilar from the training set, except for one molecule representing a known hit. The task involves ranking the activity of the molecules within clusters, hence we use mean intercluster Spearman correlation to evaluate models. To ensure that the variation in intracluster activity stems from actual differences in activity rather than random noise, we selected clusters demonstrating high variation, as detailed in Appendix B and C.

The datasets each consist of three folds. We advise using the first fold for hyperparameter selection, and then applying these hyperparameters across all folds.

Datasets are released under the MIT license. Authors bear all responsibility in case of violation of rights. Datasets are small .csv files, that is why we are going to keep them in the public GitHub repository. Reviewers can find datasets in `data` folder.

In this section, we provide further information regarding the datasets and preprocessing steps. The size and diversity of the original datasets are displayed in Table 3.

Table 3: Original datasets

| Dataset | Size | #Circles [70] (0.5) | Active fraction |
|---------|------|---------------------|-----------------|
| DRD2 (Ki) | 8482 | 837 | 0.731 |
| HIV | 41127 | 19222 | 0.035 |
| KDR (IC50) | 8826 | 791 | 0.643 |
| Sol | 2173 | 1763 | 0.216 |
| KCNH2 (IC50) | 11159 | 2128 | NA |

## A.1  Data preprocessing

We began by canonicalizing all SMILES using RDKit 2022.9.5.

For `DRD2-Hi`, `DRD2-Lo`, `KDR-Hi`, `KDR-Lo` and `KCNH2-Lo` we utilized data from the ChEMBL30 [75] database. We collected data points that measured Ki (for DRD2) and IC50 (for KCNH2 and KDR) with `confidence_score` $\geq$ 6. We selected those for which `standard_units` were in "nM". We converted `standard_value` to logarithmic scale, also known as pChembl(https://chembl.gitbook.io/chembl-interface-documentation/frequently-asked-questions/chembl-data-questions#what-is-pchembl).

For binary `DRD2-Hi` and `KDR-Hi` we binarized the data such that log activity values greater than 6 (which is < 10 muM) were designated as 1, and all others as 0. We removed any ambiguous data points (e.g. with `standard_relation` of "<" and an activity value more than 10 muM, because those could not be binarized reliably). Following this, we selected data points with identical SMILES, discarding any with differing binarized activities.

For the continuous `DRD2-Lo`, `KDR-Lo` and `KCNH2-Lo` datasets, we selected data points that had `standard_relation` of '=' and a log activity value greater than 5 but less than 9. We selected data points with identical SMILES, discarding any with activity differences greater than 1.0. For the remaining data, we took the median of each group.

---

[3]Lo-Hi benchmark: https://github.com/SteshinSS/lohi_neurips2023

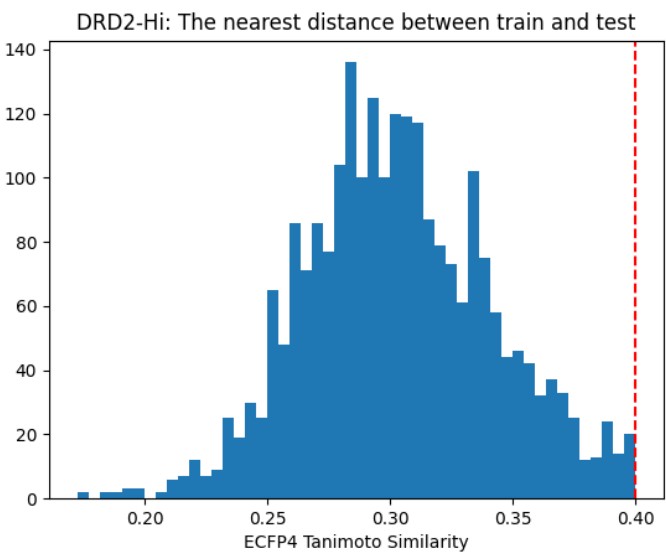

Figure 4: `DRD2-Hi`: Fold 1

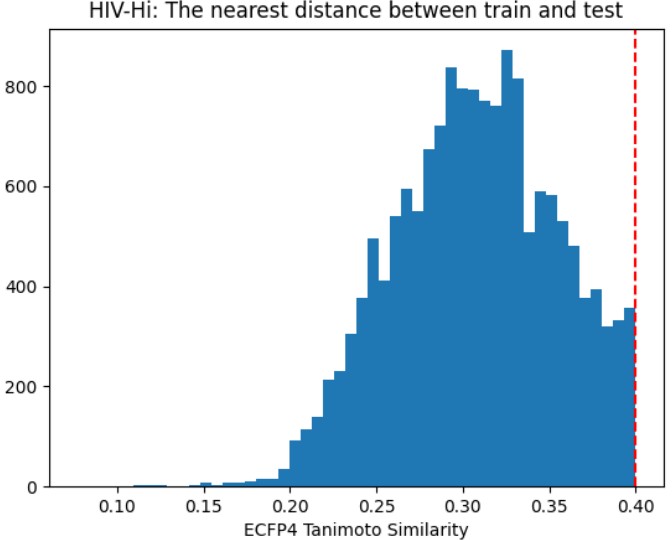

Figure 5: `HIV-Hi`: Fold 1

Table 4: Hi folds

| Dataset | Train 1 | Test 1 | Train 2 | Test 2 | Train 3 | Test 3 |
|---------|---------|--------|---------|--------|---------|--------|
| DRD2-Hi | 2385 | 1190 | 2381 | 1194 | 2384 | 1191 |
| HIV-Hi | 15696 | 7847 | 15695 | 7848 | 15695 | 7848 |
| KDR-Hi | 500 | 3116 | 500 | 3125 | 500 | 2285 |
| Sol-Hi | 1442 | 721 | 1442 | 721 | 1442 | 721 |

Table 5: Lo folds

| Dataset | Train 1 | Test 1 | Train 2 | Test 2 | Train 3 | Test 3 |
|---------|---------|--------|---------|--------|---------|--------|
| DRD2-Lo | 2206 | 267 | 2128 | 267 | 2257 | 262 |
| KCNH2-Lo | 3313 | 406 | 3313 | 406 | 3313 | 406 |
| KDR-Lo | 500 | 437 | 500 | 520 | 500 | 417 |

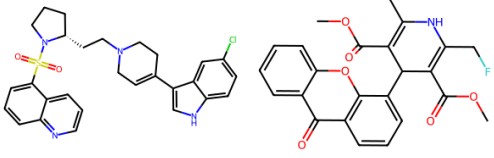

DRD2-Hi: Fold 1 train          HIV-Hi: Fold 1 train

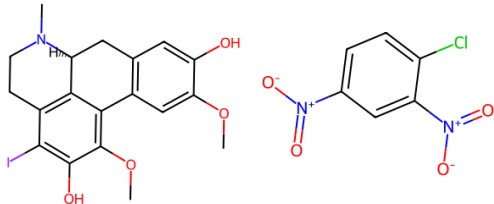

DRD2-Hi: Fold 1 test          HIV-Hi: Fold 1 test

Figure 6: The most similar pairs of molecules between train and test.

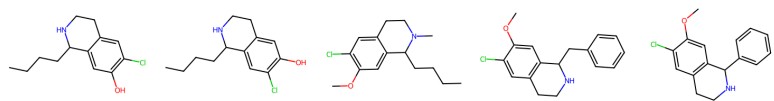

Figure 7: Example of Lo cluster in DRD2-Lo

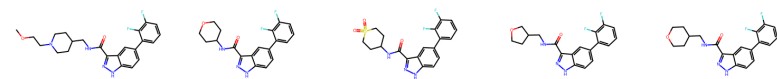

Figure 8: Example of Lo cluster in KCNH2-Lo

# B   Lo dataset is not just noise

Experimental data inherently contain noise. Consequently, selecting similar molecules may result in clusters that possess such a small variation that it could be solely attributable to experimental noise, thereby invalidating the Lo task. This potential issue underlines the importance of ascertaining that the clusters exhibit a significant signal to ensure the validity of the task.

As reported [88], the standard deviation for the same ligand-protein pair's pIC50 is $\sigma_{pIC50} \approx 0.20$ when measured in the same laboratory, and $\sigma_{pIC50} \approx 0.68$ in the ChEMBL database. In similar work [89] standard deviation for ChEMBL pKi was found to be $\sigma_{pKi} \approx 0.56$. Therefore, based on these findings, we opted to select only those clusters that displayed a standard deviation exceeding $0.70$ for pIC50 and more than $0.60$ for pKi. These selection criteria enhance the confidence in the validity of the Lo task by prioritizing clusters with significant intracluster variation.

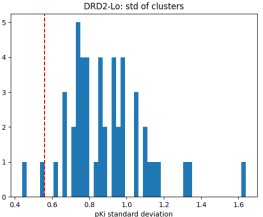 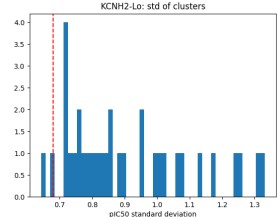 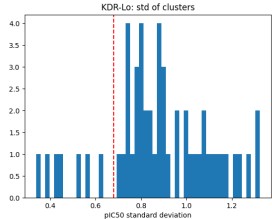

Figure 9: Within cluster variability is higher than noise standard deviation.

# C   Lo algorithm

The Python implementation can be found in `code/splits.py`.

---
**Algorithm 1** Get Lo Split

---
**Input:** List of molecular SMILES $S$, similarity threshold $t$, minimum cluster size $m$, maximum number of clusters $M$, activity values $V$, standard deviation threshold $std_t$
**Output:** List of SMILES clusters $C$, list of remaining training SMILES $train\_S$
1: **procedure** GETLOSPLIT($S, t, m, M, V, std_t$)
2:    $C, train\_S \leftarrow$ SELECTDISTINCTCLUSTERS($S, t, m, M, V, std_t$)
3:    **for** each $cluster$ in $C$ **do**
4:       Move central molecule from $cluster$ to $train\_S$
5:    **end for**
6:    **return** $C, train\_S$
7: **end procedure**

---

**Algorithm 2** Select Distinct Clusters

---

**Input:** List of molecular SMILES $S$, similarity threshold $t$, minimum cluster size $m$, maximum number of clusters $M$, activity values $V$, standard deviation threshold $std_t$
**Output:** List of SMILES clusters $C$, list of the rest training SMILES $train\_S$
1: **function** SELECTDISTINCTCLUSTERS($S, t, m, M, V, std_t$)
2:    $train\_S \leftarrow S$
3:    Initialize list $C$ as empty
4:    **while** length of $C < M$ **do**
5:        Compute fingerprints $F$ from SMILES in $train\_S$
6:        Compute total number of neighbors $N$ for each fingerprint in $F$
7:        Compute $STD$ standard deviation of $V$ of neighbors for each fingerprint in $F$
8:        Set $central\_idx$ to None
9:        Set $least\_neighbors$ to max($N$)
10:       **for** each $idx$ in $0..|train\_S|$ **do**       ▷ Find the smallest cluster that meets criteria
11:          **if** $N[idx] > m$ and $STD[idx] > std_t$ and $N[idx] < least\_neighbors$ **then**
12:             $central\_idx \leftarrow idx$
13:             $least\_neighbors \leftarrow N[idx]$
14:          **end if**
15:       **end for**
16:       **if** $central\_idx$ is None **then**       ▷ Exit if there are no more clusters that meet criteria
17:          **break**
18:       **end if**
19:       Add $central\_idx$ molecule and its neighbors to list of clusters $C$
20:       Remove the cluster and its neighbors from $train\_S$
21:    **end while**
22:    **return** $C, train\_S$
23: **end function**

---

## D   Additional benchmarks analysis

Distribution of Tanimoto Similarity between the nearest molecules between train and test.

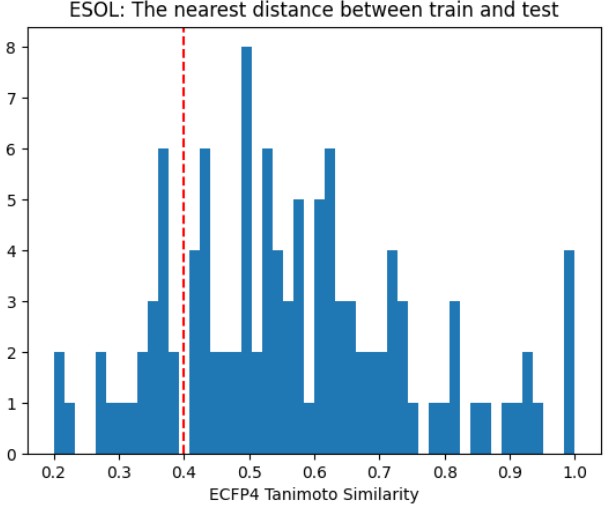

Figure 10: ESOL

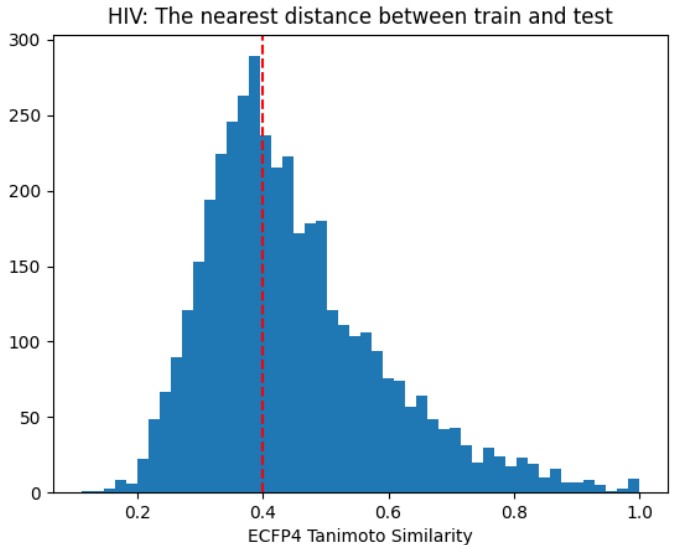

Figure 11: `HIV`

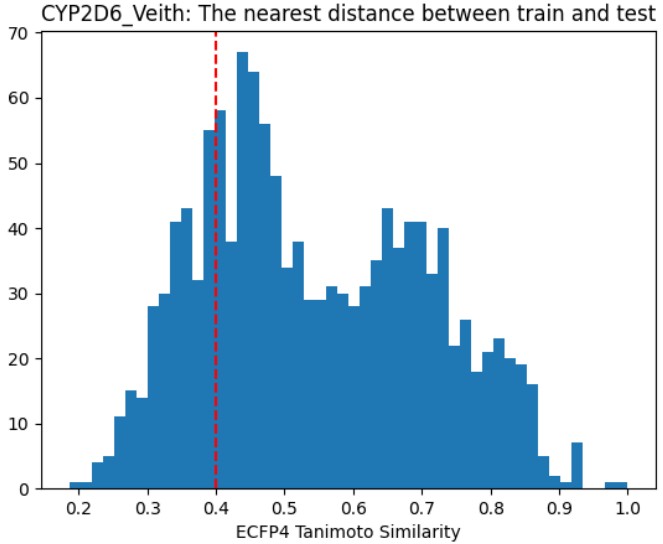

Figure 12: `TDC`

We additionally analyzed other ligand-based MoleculeNet datasets.

Table 6: Fraction of test molecules in various MoleculeNet datasets with a Tanimoto similarity >0.4 to the train set using ECFP4 fingerprints.

| Dataset | Fraction of Test Molecules Similar to Train Set |
|---|---|
| QM7 | 0.93 |
| QM8 | 0.98 |
| QM9 | 0.99 |
| FreeSolv | 0.8 |
| Lipophilicity | 0.67 |
| PCBA | >0.93 |
| MUV | 0.96 |
| BACE | 0.77 |
| Tox21 | 0.52 |
| SIDER | 0.48 |

# E   Graph coarsening algorithm

The Python implementation can be found in `code/min_vertex_k_cut.py`. We are planning to release it as a pip package.

---

**Algorithm 3** Calculate Neighbors

---

**Input:** Graph $G = (V, E)$, similarity threshold $\theta$
**Output:** List of tuples $n\_neighbors$
 1: **function** CALCULATENEIGHBORS($G, \theta$)
 2:     Initialize list $n\_neighbors$ as empty
 3:     **for** each node $v$ in $V$ **do**
 4:         Initialize $total\_neighbors$ as 0
 5:         **for** each edge $e$ incident on node $v$ **do**
 6:             **if** $e$['similarity'] $> \theta$ **then**
 7:                 $total\_neighbors \leftarrow total\_neighbors + 1$
 8:             **end if**
 9:         **end for**
10:         Append $(total\_neighbors, \text{index of } v)$ to $n\_neighbors$
11:     **end for**
12:     **return** $n\_neighbors$
13: **end function**

---

**Algorithm 4** Cluster Nodes

---

**Input:** Sorted list $n\_neighbors$, Graph $G = (V, E)$, similarity threshold $\theta$
**Output:** Cluster assignment $node\_to\_cluster$, number of clusters $total\_clusters$
 1: **function** CLUSTERNODES($n\_neighbors$, $G$, $\theta$)
 2:     Initialize array $node\_to\_cluster$ of size $|V|$ as $-1$
 3:     Initialize $total\_clusters$ as 1
 4:     **for** each tuple $(count, node)$ in $n\_neighbors$ **do**
 5:         **if** $node\_to\_cluster[node] = -1$ **then**
 6:             $node\_to\_cluster[node] \leftarrow total\_clusters$         ▷ Assign new cluster
 7:             **for** each edge $e$ incident on node $node$ **do**
 8:                 **if** $e['\text{similarity}'] > \theta$ **then**
 9:                     $adjacent\_node \leftarrow e[1]$
10:                     **if** $node\_to\_cluster[adjacent\_node] = -1$ **then**
11:                         $node\_to\_cluster[adjacent\_node] \leftarrow total\_clusters$
12:                     **end if**
13:                 **end if**
14:             **end for**
15:             $total\_clusters \leftarrow total\_clusters + 1$
16:         **end if**
17:     **end for**
18:     **return** $node\_to\_cluster$, $total\_clusters$
19: **end function**

---

**Algorithm 5** Build Coarse Graph

---

**Input:** Cluster assignment $node\_to\_cluster$, number of clusters $total\_clusters$, Graph $G = (V, E)$
**Output:** Coarsened Graph $G_{\text{coarse}}$
 1: **function** BUILDCOARSEGRAPH($node\_to\_cluster$, $total\_clusters$, $G$)
 2:     Compute $clusters\_size$, count of each unique element in $node\_to\_cluster$
 3:     Initialize $G_{\text{coarse}}$ as an empty graph
 4:     **for** $cluster$ in 0 to $total\_clusters - 1$ **do**         ▷ Add nodes
 5:         Add node $cluster$ with weight $clusters\_size[cluster]$ to $G_{\text{coarse}}$
 6:     **end for**
 7:     **for** $cluster$ in 0 to $total\_clusters - 1$ **do**         ▷ Add edges
 8:         Initialize $connected\_clusters$ as an empty set
 9:         Get nodes of $cluster$ as $this\_cluster\_indices$ where $node\_to\_cluster$ equals $cluster$
10:         **for** each $node$ in $this\_cluster\_indices$ **do**
11:             **for** each edge $e$ incident on node $node$ **do**
12:                 Add $node\_to\_cluster[e[1]]$ to $connected\_clusters$
13:             **end for**
14:         **end for**
15:         **for** each $connected\_cluster$ in $connected\_clusters$ **do**
16:             Add edge from $cluster$ to $connected\_cluster$ in $G_{\text{coarse}}$
17:         **end for**
18:     **end for**
19:     **return** $G_{\text{coarse}}$
20: **end function**

---

**Algorithm 6** Main Procedure

---

**Input:** Graph $G = (V, E)$, similarity threshold $\theta$
**Output:** Coarsened graph $G_{\text{coarse}}$
 1: **procedure** COARSEGRAPH($G$, $\theta$)
 2:     $n\_neighbors \leftarrow$ CALCULATENEIGHBORS($G$, $\theta$)
 3:     Sort $n\_neighbors$ in descending order of first element of each tuple
 4:     $node\_to\_cluster$, $total\_clusters \leftarrow$ CLUSTERNODES($n\_neighbors$, $G$, $\theta$)
 5:     $G_{\text{coarse}} \leftarrow$ BUILDCOARSEGRAPH($node\_to\_cluster$, $total\_clusters$, $G$)
 6:     **return** $G_{\text{coarse}}$
 7: **end procedure**

## F  Hi-split predicts virtual screening hit rate better than scaffold split

For effective virtual screening, predicting experimental outcomes prior to experimentation is paramount. In this study, we compare the predictive performance of the novel Hi-split approach with the traditional scaffold split method under a Hit Identification scenario. Following existing literature [10, 11, 16, 27, 28], we simulate testing on novel molecules with an ECFP4 Tanimoto similarity of $\leq 0.4$ to the training set. The dataset is partitioned using both splitting methods to form separate training and validation sets for hyperparameter selection. Hyperparameter search is performed for gradient boosting on ECFP4 fingerprints, identified as the most efficient Hi model that facilitates quick training.

After selecting the optimal hyperparameters, performance metrics are computed on the validation set. Subsequently, the model is retrained on the combined training and validation sets, and performance metrics for the hold-out test set are calculated to simulate the application of a trained model in virtual screening. The results are summarized in Table 7.

Table 7: Hi-split vs scaffold split

| Dataset | Validation | Test |
|---|---|---|
| DRD2-Hi (Hi split) | 0.603 | **0.677** |
| DRD2-Hi (Scaffold split) | 0.872 | 0.663 |
| HIV-Hi (Hi split) | 0.069 | **0.084** |
| HIV-Hi (Scaffold split) | 0.189 | 0.078 |

The Hi-split method demonstrates superior predictive performance for virtual screening hit rate compared to the scaffold split method, which is over-optimistic in the Hit Identification scenario. It also improved the test evaluation metric, although the difference is not substantial. The improved performance of the Hi-split may be attributed to the selection of more regularized models.

## G  Novelty consensus analysis

We have reproduced the work presented in [43] using binary ECFP4 fingerprints, as calculated by RDKit version 2022.9.5. The results can be found in Figure 13. For this particular work, we selected 0.40 as the novelty threshold.

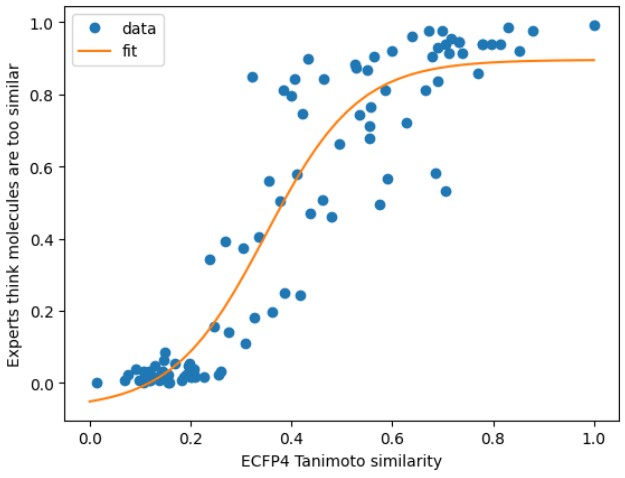

Figure 13: Sigmoid fit to [43] data

# H Hyperparameter optimization

We used random or grid search to optimize hyperparameters for all models except for the Graphormer, which was too slow for meticulous hyperparameter search. Here we provide optimization parameters and additional commentary on the training.

We utilized a single NVIDIA RTX 2070 SUPER with CUDA 11.7 and calculated binary 1024 ECFP4 and MACCS fingerprints using RDKit 2022.9.5.

## H.1 Dummy baseline

Always predicts the same constant value.

## H.2 KNN

We used `scipy.spatial.distance.jaccard` as the distance metric, as it outperformed the standard Euclidian distance in our use case. We used grid search with all combinations of parameters. For ECFP4 it was:

```
params = {
    'n_neighbors': [3, 5, 7, 10],
    'weights': ['uniform', 'distance'],
}
```

and for MACCS:

```
params = {
    'n_neighbors': [3, 5, 7, 10, 12, 15],
    'weights': ['uniform', 'distance'],
}
```

## H.3 Gradient Boosting

We used 30 iterations of random search with these parameters:

```
params = {
    'n_estimators': [10, 50, 100, 150, 200, 250, 500],
    'learning_rate': [0.01, 0.1, 0.3, 0.5, 0.7, 1.0],
    'subsample': [0.4, 0.7, 0.9, 1.0],
    'min_samples_split': [2, 3, 5, 7],
    'min_samples_leaf': [1, 3, 5],
    'max_depth': [2, 3, 4],
    'max_features': [None, 'sqrt']
}
```

## H.4 SVM

We used grid search with these parameters:

```
params = {
    'C': [0.1, 0.5, 1.0, 2.0, 5.0],
}
```

## H.5 MLP

We implemented a feed-forward neural network using Pytorch 2.0.0+cu117 and Pytorch Lightning 2.0.2. It consisted of several feed-forward layers with optional dropout layers. We used early stopping to prevent overfitting with patience 20 for the Hi tasks, and 10 for the Lo tasks. We used learning rate 0.01. We used batch size 32. We conducted 30 iterations of random search. For ECFP4 we used these parameters:

```
param_dict = {
    'layers': [
        [1024, 32, 32],
        [1024, 16, 16],
        [1024, 32],
        [1024, 8, 4],
        [1024, 4]
    ],
    'dropout': [0.0, 0.0, 0.2, 0.4, 0.6],
    'l2': [0.0, 0.0, 0.001, 0.005, 0.01],
}
```

For MACCS we used these parameters:

```
param_dict = {
    'layers': [
        [167, 32, 32],
        [167, 16, 16],
        [167, 32],
        [167, 8, 4],
        [167, 4]
    ],
    'dropout': [0.0, 0.0, 0.2, 0.4, 0.6],
    'l2': [0.0, 0.0, 0.001, 0.005, 0.01],
}
```

After the selection of the best hyperparameters, we selected a fixed number of the training epochs using early stopping. We used the same number of epochs for all the folds.

## H.6 Chemprop

We used Chemprop 1.5.2 with rdkit features. We found the evaluation metrics to be a little better with them, but it was SOTA for Hi even without them:

```
'--features_generator rdkit_2d_normalized',
'--no_features_scaling',
```

We used 20 iterations of random search with these parameters:

```
param_dict = {
    '--depth': ['3', '4', '5', '6'],
    '--dropout': ['0.0', '0.2', '0.3', '0.5', '0.7'],
    '--ffn_hidden_size': ['600', '1200', '2400', '3600'],
    '--ffn_num_layers': ['1', '2', '3'],
    '--hidden_size': ['600', '1200', '2400', '3600']
}
```

We selected the number of epochs using only the first fold. After the hyperparameters were selected, we trained the model and did not expose it to the test data. The full command for training Chemprop for HIV-Hi dataset:

```
chemprop_train --data_path data/hi/hiv/train_1.csv --dataset_type classification \
--save_dir checkpoints/hi/hiv/ \
--config_path configs/hiv_hi \
--separate_val_path data/hi/hiv/train_1.csv \
--separate_test_path data/hi/hiv/train_1.csv \
--metric 'prc-auc' \
--epochs 40 \
--features_generator rdkit_2d_normalized \
--no_features_scaling
```

For the `DRD-Hi` the best hyperparameters were:

```
{
"depth": 6,
"dropout": 0.0,
"ffn_hidden_size": 2400,
"ffn_num_layers": 1,
"hidden_size": 2400
}
```

For the `HIV-Hi` the best hyperparameters were:

```
{
"depth": 6,
"dropout": 0.2,
"ffn_hidden_size": 3600,
"ffn_num_layers": 2,
"hidden_size": 3600
}
```

### H.7 Graphormer

We used Graphormer with the last commit 77f436db46fb9013121289db670d1a763f264153. We applied two fixes, that we found in issues `https://github.com/microsoft/Graphormer/issues/158#issuecomment-1500311589` and `https://github.com/microsoft/Graphormer/issues/130#issuecomment-1207316808` that solved our problems. However, we set up an in-house Graphormer some time ago and currently, it cannot be reinstalled from scratch due to multiple broken dependencies.

We modified code to calculate and track PR AUC metrics, to add our datasets, and to evaluate trained models. We manually optimized the hyperparameters over approximately 10 iterations. We found Graphormer to be inferior to Chemprop, which is consistent with our previous experience with different datasets.

We faced numerous technical difficulties in executing and modifying Graphormer [81, 82] due to improper dependency pinning by the authors. We found the training to be slow, which limited our ability to optimize hyperparameters. Because of technical difficulties, we decided not to test it for the Lo task.

### H.8 `HIV-Hi` balance

`HIV-Hi` is a highly unbalanced binary classification problem with only 3% of positive examples. Due to this imbalance, we experimented with weighted options of classical ML algorithms and manually resampled positive examples for neural networks.

# I Spearman distribution

The test set of the Lo datasets is composed of molecular clusters. To evaluate the models, the Spearman correlation coefficient is calculated within each cluster, comparing the actual activity values to the predicted ones. The final Lo metric is the average of the Spearman coefficients across all clusters.

In the following, we present a histogram of Spearman coefficients for the best models across various datasets. Note that the KDR-Lo dataset is more challenging than both DRD2-Lo and KCNH2-Lo.

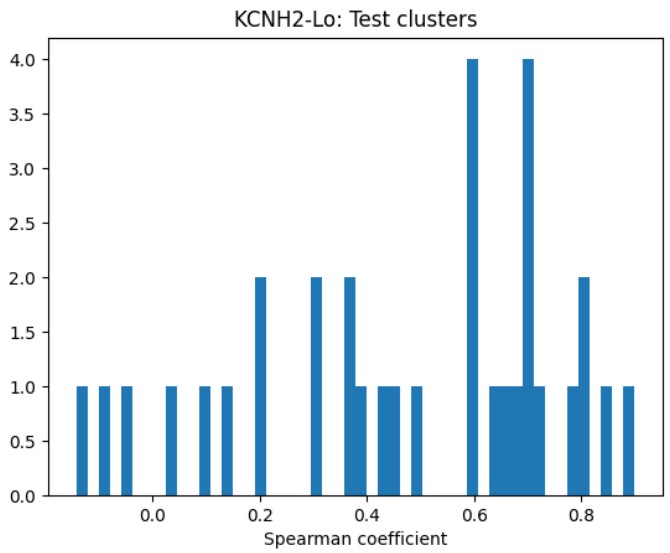

Figure 14: KCNH2-Lo Spearman coefficient distribution for SVM-ECFP4

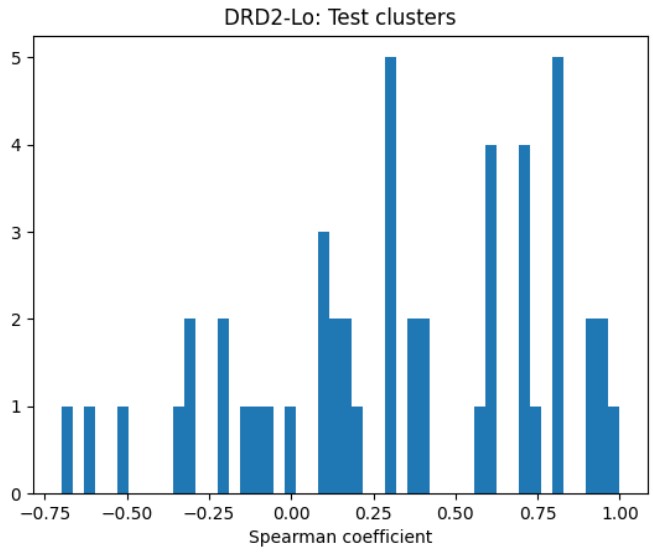

Figure 15: DRD2-Lo Spearman coefficient distribution for SVM-ECFP4

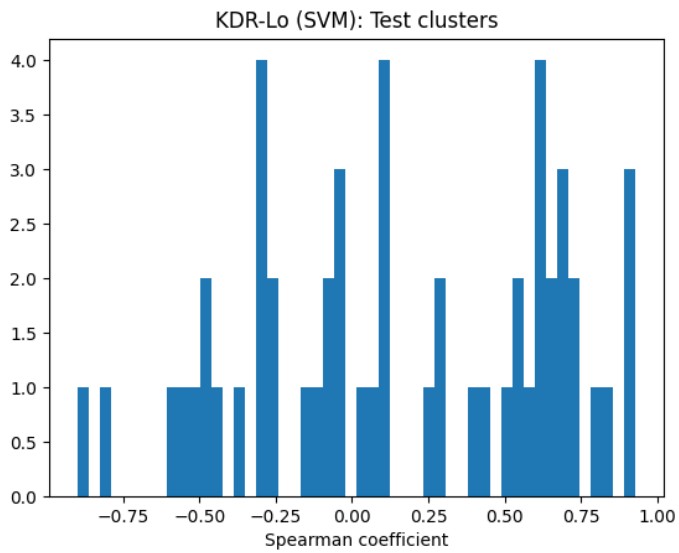

Figure 16: KDR-Lo Spearman coefficient distribution for SVM-ECFP4

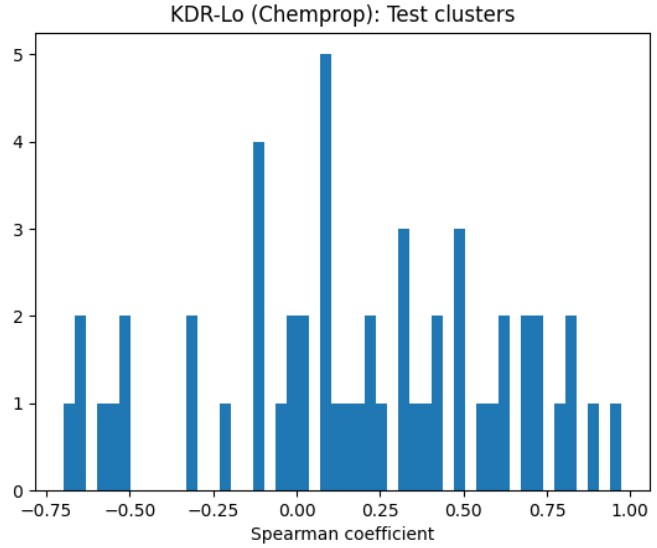

Figure 17: KDR-Lo Spearman coefficient distribution for Chemprop

