# OpenReview forum: "Lo-Hi: Practical ML Drug Discovery Benchmark"
_NeurIPS.cc/2023/Track/Datasets_and_Benchmarks — NeurIPS 2023 Datasets and Benchmarks Poster_

### Official Review · Reviewer_igzy · 2023-06-21
**Important study pointing out major flaws in current drug discovery benchmarks and introduction of an improved dataset**

**Rating:** 9
**Confidence:** 3

**Strengths:**

As above the paper tackles a very important topic for multiple research communities. The introductory logic is compelling, clear and well-written for the difference between lead optimization and hit identification, how the drug-discovery process works and interacts (or fails to interact) with frontier machine learning and so on. The supplementary material is extensive and shows that the author have made considerable efforts to provide clear evidence for claims in the paper, e.g. with substantial analysis into hyperparameters and different architectures, etc..

**Additional Feedback:**

As above, mostly just provide more insight into what the next steps should be in terms of expanding or improving drug discovery benchmarks along the vision presented in this work.

**Clarity:**

There are small English errors throughout materials, e.g. " Lo dataset is not a noise" in the supplement as a section title, but one round of a spell/grammar tool or checker should help this. Otherwise the logic and analysis in the paper is clear.

**Correctness:**

To the best of my knowledge, the authors have used the correct and appropriate frontier ML/chem interface methods properly in their work (Rdkit, SMILES, Tanimoto similarity, etc.). I cannot find any errors to mention.

**Documentation:**

In addition to raw code, the authors have provided helpful pseudocode and clustering examples in the supplementary information. The documentation overall seems extensive and helpful to anyone who would wish to use their data and code.

**Limitations:**

The mentioned limitations almost exclusively focus on ML issues such as hyperparmater tuning and various architectural difficulties, which is certainly helpful. This is a dataset submission though, so comments on data limitations would also be very helpful. Currently the authors have written:

"We selected datasets that represent realistic drug discovery problems, contain substantial amounts of qualitative data, and cover a diverse chemical space..."

which is great, but no dataset is absolutely perfect. Could the authors discuss any important limitations of the dataset for future work to be aware of or consider improving?

**Opportunities For Improvement:**

The claims about issues with current benchmarks are so extreme, I just wonder how there could be no discussion about them before this work, especially in the ML community. Could the authors improve the literature review with any papers that have pointed out similar flaws in drug discovery benchmark work before, specifically on the ML side? Most references seem to be in the biomedical domain instead. Else confirm they have comprehensively survived the relevant ML literature and verified minimal-to-no prior awareness of the major problems discussed?

The Figure 15 and 16 Spearman plots are presented without any further commentary or analysis. Could the authors help interpret what the distributions mean and how they should be understood in the context of assessing the current dataset quality? Similar for other figures in the supplement.

Otherwise the authors seem to have done substantial work in justifying their claims in terms of logic and analysis.

**Relation To Prior Work:**

As above the prior lit discussion is acceptable in both the ML side and chemistry/biomedical domain knowledge side, but the references lean much more towards biomedical citations so it is not entirely clear that the authors have accurately depicted the proposed lack of awareness of major errors in drug discovery benchmarks in ML research.

**Summary And Contributions:**

If the claims are accurate, this paper is potentially very high impact and useful for the ML-biomedical boundary community, as it points out both major flaws and clear roadmaps for solutions in the important area of drug discovery datasets in a clearly written way, demonstrating substantial ML and domain knowledge.

Example quotes below related to the potential impact:

"We demonstrate that none of the modern benchmarks correspond to realistic conditions, thus raising questions about their suitability for evaluating practical machine learning models... which is why we propose four new practical datasets — two for each task — that better imitate real-life drug discovery scenarios"

"In the Hi scenario, the training and test sets are significantly different to simulate the search for new molecules, making the Hi scenario interpretable as an Out-Of-Distribution (OOD) task. Although OOD benchmarks already exist for machine learning-based drug discovery, they do not align with practical drug discovery applications. DrugOOD [20] is a drug discovery benchmark dedicated to Out-Of-Distribution prediction. The authors accurately observe that "In the field of AI-aided drug discovery, the problem of distribution shift [...] is ubiquitous", and therefore suggest assay-based, scaffold-based, or molecular size-based partitions to create a distribution shift. However, their benchmark does not correspond to practical drug discovery, as it involves predicting average activity for all ChEMBL targets simultaneously in ligand-based drug discovery datasets. This approach is impractical because, in reality, researchers are not interested in average activity across ChEMBL targets, but rather in activity on a specific target. GOOD [21] is a benchmark explicitly designed to separate covariate and concept shifts."

"...We show that modern benchmarks mix these two scenarios, which is why it remains unclear how effectively models can generalize to truly novel molecules and how proficiently they can distinguish minor modifications, thereby guiding molecular optimization. Furthermore, we untangle these scenarios using a novel benchmark..."

---

> ### Author Response · Authors · 2023-08-17
>
> Thank you for your review and kind words.
>
> Indeed, as you've pointed out, our paper aims to bridge a crucial gap, and we genuinely believe it has the potential to be a high-impact contribution. Furthermore, we did our utmost to ensure the clarity of the paper, so it's reassuring to learn that this resonated in your assessment.
>
> Addressing your valuable suggestions:
>
> 1. **Supplementary Figures Commentary:** Upon your recommendation, we have now added more detailed commentaries to Figures 14 - 17, elucidating the underlying patterns and their significance in the broader context of dataset quality evaluation. Similar enhancements have been applied to other supplementary figures as well.
> 2. **Dataset Limitations & Future Work:** We acknowledge that no dataset is without its imperfections. Based on your feedback, we have appended a section detailing potential limitations inherent to our dataset and have proposed directions for future works to refine these aspects.
> 3. **Clarity and Language Corrections:** We have corrected “Lo dataset is not a noise”, thanks.
>
> **Literature Review.**
>
> > Could the authors improve the literature review with any papers that have pointed out similar flaws in drug discovery benchmark work before specifically on the ML side? Most references seem to be in the biomedical domain instead. Else confirm they have comprehensively survived the relevant ML literature and verified minimal-to-no prior awareness of the major problems discussed? ****
>
> The short answer is — yes. We have thoroughly reviewed the relevant ML literature and found that there's minimal awareness of the problem.
>
> For a more extended answer, which you might find interesting:
>
> We wouldn’t say that we focus solely on the biomedical domain. Instead, our primary interest lies in ML drug discovery with practical validation, as seen in [1-4]. Given that the main objective of ML in drug discovery is to find new drugs, we value practical works more than those that simply beat benchmarks. Additionally, the authors of these works are well-known for their theoretical contributions in ML for drug discovery
>
> Based on the cited works, there exists a misalignment between training objectives and their real-world applications, which we aim to bridge using our dataset splits.
>
> **Lo**
>
> There are approximately 200 studies on molecular generative models. Some employ a predictive model to guide optimization in a step-by-step manner. I've extensively reviewed the literature on molecular generation, yet I've not encountered anyone testing whether their predictive model can discern such minor modifications.
>
> **Hi**
>
> Conversely, in Molecular Property Prediction, there are papers that apply models to ECFP4-dissimilar molecules. Examples include practical validation works like [1-3] and theoretical studies such as [5, NeurIPS 2022] and [6, ICLR 2023 ML Drug Discovery workshop].
>
> We discuss molecular generative models in the context of the Lo task. However, a recent study [7] introduced a generative model that produces molecules in the Hi regime, exhibiting a Tanimoto similarity of <0.4 to the training set. We stumbled upon this post-submission but have since included the reference.
>
> This closely relates to my favorite paper on molecular generative models [8]. It is primarily renowned for its new SOTA model "AddCarbon", which simply inserts "C" at a random position in SMILES. However, the second part of the paper demonstrates that goal-directed generative models with a predictive model are highly incentivized to replicate the training set of the predictive model. [7] addresses this issue.
>
>
>
> While some recognize the significance of novelty, I haven't come across discussions on better dataset splits, measuring novelty (in ML drug discovery), or standardized benchmarks for novel molecules. Our work aims to fill this void.
>
> [1] Stokes, Jonathan M., et al. "A deep learning approach to antibiotic discovery." *Cell* 180.4 (2020): 688-702.
>
> [2] Wong, Felix, et al. "Discovering small-molecule senolytics with deep neural networks." *Nature Aging* (2023): 1-17.
>
> [3] Liu, Gary, et al. "Deep learning-guided discovery of an antibiotic targeting Acinetobacter baumannii." *Nature Chemical Biology* (2023): 1-9.
>
> [4] Godinez, William J., et al. "Design of potent antimalarials with generative chemistry." *Nature Machine Intelligence* 4.2 (2022): 180-186.
>
> [5] Lu, Wei, et al. "Tankbind: Trigonometry-aware neural networks for drug-protein binding structure prediction." *Advances in neural information processing systems* 35 (2022): 7236-7249.
>
> [6] Owoyemi, Joshua, and Nazim Medzhidov. "SmilesFormer: Language Model for Molecular Design." *ICLR 2023-Machine Learning for Drug Discovery workshop*. 2023.
>
> [7] Lee, Seul, Jaehyeong Jo, and Sung Ju Hwang. "Exploring chemical space with score-based out-of-distribution generation." *International Conference on Machine Learning*. PMLR, 2023.
>
> [8] Renz, Philipp, et al. "On failure modes in molecule generation and optimization."

---

> > ### Comment · Reviewer_igzy · 2023-08-18
> > **Response is satisfactory**
> >
> > Thank you, these response comments in conjunction with the responses to the other reviewers have satisfied my concerns.

---

### Official Review · Reviewer_ZG59 · 2023-07-19
**High quality paper with a refreshingly accurate perspective on drug discovery benchmarks**

**Rating:** 7
**Confidence:** 4
**Correctness:** As far as I can tell, everything is c…
**Clarity:** Yes, it is very clearly written!

**Strengths:**

- Very good motivation for the benchmark: the authors clearly understand how molecular property prediction models will be deployed and design the benchmark accordingly. The design makes up for deficiencies in other benchmarks (chiefly data splitting). It is refreshing to see a paper propose a benchmark around real-world drug discovery practices, instead of taking a problem setting from computer vision / NLP and trying to make a version with molecules!
- Very clear writing throughout. The paper was a pleasure to read.
- Case studies on other datasets in section 5 were very informative. I think this analysis like this should be more common in D&B submissions. A lot of D&B submissions are very quick to dismiss other datasets based on the _intention_ of the creators rather than the contents of the dataset (e.g. "MoleculeNet was not designed with this kind of splitting in mind, so our dataset must be important."). I appreciate that you actually showed this.
- Experiments in section 7 were well-done and informative. I especially appreciate the extensive hyperparameter tuning done by the authors (many submission just use the default values, making the results meaningless in my opinion).

**Additional Feedback:**

Overall great paper, thanks for writing it.

One thing I commend you for is your analysis of DrugOOD. The dataset is essentially noise and it looks like this is why it got rejected from D&B last year. I was surprised to see that it ended up getting published. While I understand why you cited it in your related work (because it is nominally related), I think it would be better to simply ignore this paper (not cite it or discuss it) so it can be forgotten as quickly as possible...

**Documentation:**

The submission is pretty detailed, however the code link posted by the authors seems broken...

**Ethics:**

No issues.

**Limitations:**

The authors very nicely describe the limitations of their work. I think the discussion in section 9 and elsewhere is more than adequate.

**Opportunities For Improvement:**

I did not see any "mistakes" or errors in the submission, and think it is good as is. However, I do have some suggestions to make the work even better:

- Test a variety of dataset sizes: the current focus is on larger datasets, but many datasets in drug discovery are small (<1000 measurements). It would be good to extend Table 2 into several sub-benchmarks for different dataset sizes. I suspect GNNs would perform worse on smaller datasets. This would give valuable guidance to practitioners looking for the best model for their dataset.
- Obviously creating more datasets would be good: could the same procedure be applied to all datasets in MoleculeNet / TDC? That would make the benchmark much more comprehensive.
- Although the focus here is on models which make point predictions, there is a clear advantage to models which also provide uncertainty estimates (especially for out of distribution prediction, which is essentially the "hi" task). In real life it would be useful to distinguish between confident and non-confident positive predictions. It would be good to test some methods which do this out of the box (e.g. random forest, Gaussian processes) and perhaps also quantify the performance of such approaches? It would be interesting to see whether these methods actually perform better in practice.

**Relation To Prior Work:**

I think the authors have adequately discussed prior work, although I would have appreciated a longer explanation of the differences in coverage between their "lo" benchmark and previous activity cliff benchmarks.

**Summary And Contributions:**

This paper proposes a 2 benchmarks for drug discovery modelled after two separate tasks in real-world drug discovery: hit identification (predicting drug activity for novel / structurally dissimilar molecules) and lead optimization (proposing variants of hit molecules with better property scores). To create the datasets for hit identification they propose a novel graph splitting algorithm (to solve the balanced vertex k-Cut problem). Experimentally they find that GNNs perform better at hit identification, while SVM performs better for lead optimization.

---

> ### Author Response · Authors · 2023-08-17
>
> Thank you for your comprehensive and constructive feedback on our paper. We deeply appreciate the time and effort you invested in reviewing our submission.
>
> We are gratified to learn that our motivation for the benchmark resonated with you. We consistently aimed for clarity in our writing and are pleased to hear that our paper provided an engaging read.
>
> In response to your valuable suggestions (as well as feedback from other reviewers), we have incorporated small datasets, KDR-Hi and KDR-Lo, with only 500 molecules in the training set. Furthermore, we've highlighted in our future work section the importance of developing small drug discovery benchmarks akin to FS-Mol, but with an emphasis on Hi generalization.
>
> To encompass more diverse tasks, we've also introduced a new solubility dataset, Sol-Hi.
>
> Regarding the comparison between Lo and Activity cliffs prediction, we considered the important work by [1]. In their pursuit of new antimalarial drugs, the authors trained a modified JT-VAE and subsequently developed a predictor from the latent space, a-la Gómez-Bombarelli's seminal 2017 paper. They embedded existing drugs and aimed to optimize them in the latent space using the predictor. Although they presented their findings in a positive light, it seems to us that they achieved a somewhat negative result. Given that the authors did not evaluate how effectively their predictive model discerned minor modifications, there's a possibility that their model was not sensitive to such small modifications.
>
> This instance underscores a practical scenario where merely detecting activity cliffs is insufficient. The authors aimed not just to identify activity cliffs, but to traverse the latent space to enhance molecular properties.
>
> Lastly, we appreciate your pointing out the broken link. We apologize for this oversight and have now uploaded a new version to OpenReview. This should remain accessible throughout the review process.
>
> [1] Godinez, William J., et al. "Design of potent antimalarials with generative chemistry." *Nature Machine Intelligence* 4.2 (2022): 180-186.

---

> > ### Comment · Reviewer_ZG59 · 2023-08-28
> > **Changes appreciated, will keep score**
> >
> > Thanks for responding to my review. I think the changes have made the paper better, but since I was originally happy with the paper I will keep my score.
> >
> > I notice myself and 1 other reviewer recommend acceptance, while 2 other reviewers recommend rejection because they want to see more things added to the paper. I agree that their suggestions would improve the paper but I do not think that excluding these things is good grounds for rejection. I will start a discussion with the other reviewers on this (so far there has not been a discussion).

---

### Official Review · Reviewer_H78M · 2023-07-21
**Lo-Hi: Practical ML Drug Discovery Benchmark**

**Rating:** 4
**Confidence:** 3

**Strengths:**

1. The paper is relevant and timely, considering the increasing importance of machine learning in drug discovery.
2. It introduces a novel benchmark that is grounded in real-world drug discovery processes.
3. The development of the molecular splitting algorithm is a noteworthy contribution.
4. The comprehensive testing of ML models under practical settings provides valuable insights.

**Additional Feedback:**

The authors have made considerable strides in developing a practical benchmark for ML drug discovery, which is an important and timely contribution to the field. The focus on real-world tasks, such as Lead Optimization and Hit Identification, is commendable and demonstrates a clear understanding of the challenges involved in drug discovery.

The proposal to expand the target proteins beyond DRD2 and HIV is not a critique but an encouragement to broaden the impact of your work. Given the diversity of targets in drug discovery, incorporating a wider spectrum could make your benchmark even more beneficial for the community! :)

**Clarity:**

The paper is well-written and presents complex ideas in a clear manner.



**Correctness:**

The impact of outliers, especially in cases where the molecule seems entirely implausible but the model confidently predicts a positive outcome, is not explored in the manuscript. This oversight may limit the benchmark's utility in assessing model robustness and reliability, as handling such outliers is a common challenge in practical applications. The authors could consider addressing this as a separate task within their benchmark to enhance its real-world relevance and applicability.

**Documentation:**

The authors provide sufficient detail to support reproducibility. However, they could provide more information on the specific data/splits used for the benchmark.

**Ethics:**

There do not appear to be any significant ethical concerns with the submission.

**Limitations:**

1. The authors' focus on two target proteins, DRD2 and HIV, while important, may not fully reflect the complexity and diversity of practical drug discovery scenarios. Despite the authors' commendable effort in designing the benchmark tasks, the limited range of target proteins may not capture the real-world challenges and intricacies of applying machine learning in drug discovery. Expanding the scope to include a broader array of targets would likely provide a more comprehensive and realistic benchmarking environment.
2. The impact of outliers, especially in cases where the molecule seems entirely implausible but the model confidently predicts a positive outcome, is not explored in the manuscript. This oversight may limit the benchmark's utility in assessing model robustness and reliability, as handling such outliers is a common challenge in practical applications. The authors could consider addressing this as a separate task within their benchmark to enhance its real-world relevance and applicability.

**Opportunities For Improvement:**

1. While the authors have made an earnest effort in designing the benchmark tasks around two key targets, namely DRD2 and HIV, this review suggests that the scope of the tasks may need to be broadened. Drug discovery is a complex and diverse field where target proteins are numerous and varied. Restricting the benchmark to only two proteins might not entirely capture the practical challenges and nuances in employing machine learning for drug discovery.
Therefore, this review recommends that the authors consider expanding their benchmark tasks to include a more diverse array of targets. Resources like BindingDB, which offers comprehensive data on protein-small molecule interactions along with their associated binding affinities, can be utilized for this purpose. By automating the curation process over the entire database, the authors could incorporate a wider spectrum of targets into their benchmark, thereby enhancing its practical relevance and applicability.'

2. From a new user's perspective, It would be great if authors could provide instructions within the manuscript. For example, is a validation set ever used for performing the training? how is it split? Do the authors provide pre-existing splits? Incorporating these will help a new user.

3. It would be beneficial if the authors could investigate the impact of outliers. Specifically, situations where the molecule appears entirely unfeasible, yet the model confidently predicts a positive outcome. From a practical standpoint, this reviewer has encountered many such instances and would not be surprised if similar occurrences are happening with the authors' models. Could this issue be addressed as a distinct task?
Two references that may be useful in this context are: "STONED-SELFIES" (https://pubs.rsc.org/en/content/articlehtml/2021/sc/d1sc00231g) for generating outliers, and "Model agnostic generation of counterfactual explanations for molecules", which discusses how counterfactuals can be produced.

**Relation To Prior Work:**

The authors clearly discuss how this work differs from previous contributions, particularly the limitations of existing benchmarks for drug discovery.

**Summary And Contributions:**

The manuscript presents a novel benchmark called "Lo-Hi" for drug discovery using machine learning. This benchmark includes two tasks: Lead Optimization (Lo) and Hit Identification (Hi). The authors propose a unique molecular splitting algorithm, which is designed to address the Balanced Vertex Minimum k-Cut problem. The work presents a critique of existing benchmarks and provides comprehensive tests on classic and state-of-the-art ML models under practical settings.

---

> ### Author Response · Authors · 2023-08-17
>
> Thank you for your comprehensive review and constructive suggestions. We genuinely appreciate the time and effort you have dedicated to improving our manuscript.
>
> ### Dataset Diversity
>
> We acknowledge the importance of diversifying our benchmark to provide a more holistic view of the drug discovery process through machine learning. Based on your recommendations, we have made several key updates:
>
> We have introduced a new KDR-Hi and KDR-Lo datasets. The choice of KDR (kinase target) complements our existing GPCR DRD2 dataset, providing more diverse molecular targets for assessment.
>
> Additionally, recognizing the importance of drug solubility, we have integrated the Sol-Hi dataset. Solubility is an essential property that impacts the pharmacokinetics and, ultimately, the success of a therapeutic compound.
>
> We'd like to highlight that our benchmark was not solely restricted to DRD2 and HIV; it also encompasses KCNH2, a target of cardiotoxicity assays. This inclusion underscores our intention to address the multifaceted challenges in drug discovery.
>
> Our revised benchmark now features seven distinct datasets, each subjected to a three-fold validation process. We not only preprocessed datasets but also established firm baselines to ensure the non-triviality of the next-gen SOTA models.
>
> ### Documentation
>
> We concur with the necessity of ensuring that our datasets and methodologies are user-friendly and can be readily employed without an exhaustive study of the manuscript. To this end, following the NeurIPS guidelines, our appendix begins with a succinct yet comprehensive description of the benchmark. Since our repository will be located on GitHub, we also provided a simple description in the README file within the repository. We believe this README should suffice for evaluating models, but we are prepared to modify it if any misunderstandings arise.
>
> ### Feasibility Task
>
> We fully agree on the importance of synthesizability for generated molecules. It's essential that such molecules can be practically produced in the real world. In the field of molecular generative models, synthesizability is a paramount yet underdeveloped challenge. While creating a good benchmark for this is crucial, it's a very complex task. We believe it merits a separate dedicated paper. However, we're confident that our Hi/Lo paradigm will provide insights for developing synthesizability-focused benchmarks, ensuring new models' generalizability.
>
> Once again, thank you for your thoughtful insights. Your feedback has been instrumental in enhancing the depth and breadth of our work. We hope that these improvements amplify our manuscript's contributions to the ongoing discourse on ML-driven drug discovery.

---

> > ### Comment · Reviewer_H78M · 2023-08-22
> >
> > I have reviewed the updated manuscript and the authors' responses to my initial comments. While the authors have made some effort to address the issues raised, not all of my comments have been sufficiently dealt with.
> >
> > Points Addressed:
> > - Dataset Diversity: I commend the authors for acknowledging the need to diversify the benchmark and for their effort in adding new datasets focused on different molecular targets. This is a move in the right direction.
> > - Documentation: The effort to make the dataset and methodology more accessible through a README file in the GitHub repository is appreciated.
> >
> > Points Not Addressed:
> > - Broadening the Scope: Despite adding new datasets, the scope of the benchmark still seems narrow. My initial recommendation was for a more automated curation process that could incorporate a broader array of targets from resources like BindingDB. This concern remains unaddressed.
> > - Outliers and Feasibility Task: My suggestion to investigate the impact of outliers and possibly introduce a distinct task to handle them has been acknowledged but not acted upon. While the authors consider this an important aspect, they have deferred it to future work. Given that this is a practical benchmark, the absence of this aspect could limit the benchmark's utility in real-world applications.
> > - New User's Perspective: The manuscript still lacks clear instructions for a new user, particularly regarding how the training-validation split is done. While a README is helpful, having this information in the paper itself would add value.
> >
> > While the authors have made some strides to improve the manuscript, not all the concerns raised in the initial review have been adequately addressed. These remaining issues limit the paper’s impact and therefore, my final recommendation is to not change my score.
> >
> > However, I believe that the authors are on the right path, and I encourage them to continue refining the benchmark to make it more comprehensive and relevant for practical applications in ML-driven drug discovery.
> >
> > Additional clarification point :
> > In Figure 3, it is observed that the ESOL train/test and HIV train/test molecules are exactly the same. While it is understandable that they might be similar given the overlap in molecular structure for different tasks, them being exactly the same raises concerns. This similarity could suggest issues in the dataset preparation or validation methodology, and it warrants clarification from the authors (the reviewer missed this in the original review)

---

> > > ### Author Response · Authors · 2023-08-23
> > >
> > > We appreciate your feedback. We are exploring avenues to enhance our paper and would like further elaboration on certain points.
> > >
> > > ### ESOL duplicates
> > >
> > > You are absolutely right in highlighting the concern of having identical molecules between the training and test datasets, which indeed suggests a methodological flaw in the dataset. However, it's noteworthy that MoleculeNet is the current go-to benchmark in ML for drug discovery. It's essential to underscore that we precisely followed the procedure for extracting the ESOL dataset. This is the same dataset that has been tested against numerous ML models. You can verify our process at `lohi/notebooks/paper/01_moleculenet_analysis.ipynb`.
> > >
> > > ### Broadening the scope
> > >
> > > In line with your suggestions, we've broadened our benchmark to include seven datasets. These represent a wide range of drug discovery tasks such as GPCR activity, kinase activity, phenotypic assays on HIV inhibition, cardiotoxicity, and solubility. You've proposed the inclusion of even more tasks from BindingDB.
> > >
> > > While we concur that integrating more datasets would make for a comprehensive evaluation of models, we also recognize the impracticality of benchmarking future models against every possible task. Each additional dataset requires users to dedicate more time for hyperparameter tuning and a three-fold evaluation. This poses a challenge in balancing breadth and feasibility. Would you be able to specify tasks within BindingDB that might present a new evaluative perspective, thus justifying its inclusion?
> > >
> > > ### Outliers and Feasibility Task
> > >
> > > If we've understood you correctly, you're suggesting the addition of a feasibility benchmark. This would test a model's ability to differentiate between realistic and implausible, non-synthesizable molecules. You seem to agree with our understanding, and both you and we concur that it is a very important task.
> > >
> > > Our hesitation in adding a synthesizability dataset is rooted in the absence of a consensus within the community on how to measure synthesizability. Various methods exist, from measuring feasibility based on heuristic-derived features or fragments, to regression toward expert scores, or even counting the reaction steps needed to produce a molecule. Which particular measure are you leaning towards? Why specifically this one?
> > >
> > > The most widely recognized synthesizability measure seems to be the SA score by Ertl & Schuffenhauer (2009). Are you suggesting that we incorporate their dataset into our benchmark with new splits? We're hesitant, given that their primary dataset contains only 40 molecules.
> > >
> > > Or do you suggest a more recent, yet not widely-accepted, synthesizability measure? Which one is it, and why does it seem so convincing to you that you'd like to see researchers compete to improve predictions based on that measure?
> > >
> > > We acknowledge that we might not be fully versed in every aspect of synthesizability. If there's anything we've overlooked or misunderstood, could you please clarify?
> > >
> > > ### New User’s Perspective
> > >
> > > Thank you for pointing it out. We've included additional details about dataset splitting in the paper. Kindly refer to lines 257-261 for Hi datasets and lines 287-288 for Lo datasets. If there are any other points of ambiguity, please let us know.

---

> > > > ### Comment · Reviewer_ZG59 · 2023-08-28
> > > > **Feasibility task probably not possible in general**
> > > >
> > > > I agree with the authors that there is no clear way to identify feasible/synthesizable molecules, and would further add that I think synthesizability is not a well-defined concept. However, I think it should still be possible to make a feasibility task with some specific molecules hand-chosen to be non-druglike or non-synthesizable. For example, very large molecules, unstable molecules, etc. If the authors want to create a benchmark involving feasibility that is my suggestion.

---

> > > > > ### Comment · Reviewer_igzy · 2023-08-28
> > > > > **Re: feasibility task and this discussion**
> > > > >
> > > > > In terms of whether or not the paper is accepted, the authors have done a commendable job in scope and diversity of the datasets and improving usability I think given how complex the topic is and given the schedule of the review timeline. If there is not adequate time to add in a new feasibility dataset or study, which is possible given the remaining ambiguity in the follow-up comments and the current date, I suggest having the authors just outline their best plan sketch of how one could do it in follow-up work (with any important caveats) in the Next Steps/Discussion section along the lines of their latest response to the reviewer. If the authors think there is time, then I am in agreement with ZG59.

---

> > > > > > ### Author Response · Authors · 2023-08-31
> > > > > > **New paragraph for the Future Work**
> > > > > >
> > > > > > Thank you for your support, ZG59 and igzy.
> > > > > >
> > > > > > Given the tight timeline and the intricacy surrounding the feasibility task, it's infeasible to incorporate a robust solution at this stage. While implementing Lo/Hi splits on a specific dataset is straightforward, selecting an appropriate dataset worthy of benchmark status is the real challenge. To address this, we will incorporate the following paragraph into the "Future Work" section of the camera-ready version:
> > > > > >
> > > > > > “A major ongoing challenge in molecular generative models is ensuring synthesizability, meaning that generated molecules can be made in the real world. Hi splits can help test the generalizability of synthesizability models. But, it's important to remember that Lo/Hi splits assume similar molecules have similar properties. While this holds for physico-chemical attributes, this premise remains to be validated in the context of feasibility measures.”

---

### Official Review · Reviewer_jiMQ · 2023-07-21
**Review of Lo-Hi: Practical ML Drug Discovery Benchmark**

**Rating:** 3
**Confidence:** 4

**Strengths:**

## Emphasis on testing models under representative conditions
The paper underscores the significance of assessing machine learning models within contexts analogous to their projected deployment. I find this approach compelling and believe it could lead to beneficial subsequent studies.


**Additional Feedback:**

None

**Clarity:**

The language used in this article is casual, and some statements lack the rigour expected of a scientific paper. See above for details.

**Correctness:**

## Criticism about other benchmarks and statement of this benchmark
Although this study criticizes other benchmarks for being "too dissimilar from real-world model applications," the context of this study also falls short of closely mirroring practical applications. A number of crucial facets of real drug discovery, such as data noise, data imbalance, data scarcity, and the multi-objective character of the drug, are overlooked in this benchmark. Additionally, the primary contribution of this benchmark is a method for splitting the dataset, which, on the whole, does not substantially increase the benchmark's usefulness compared to others.

## About the drug discovery pipeline
The author delineates the processes of hit-identification and lead optimization in drug discovery, to provide context for the study. However, I find myself disagreeing with some of their assertions. During hit identification, the pursuit of novelty implies that the objective is not simply to arrive at a patented candidate; thus, it's a standard practice to remove any structures already patented from the design pool. However, when the bioactivity data of patented molecules is accessible, integrating it into the training dataset is harmless and could be beneficial. Furthermore, a typical drug discovery campaign often commences with the testing of all available in-stock molecules using a biological assay to accumulate data. Thereafter, this data can be utilized to train the model to screen a larger, virtual space. Whether or not hits are found during this initial screening, the training data for hit identification may still include some hit molecules.

## About goal-directed molecular generation
In lines 53-54, the author draws upon a quote from Gao et al.’s study to substantiate the claim that predictive models pose as the primary hurdle in goal-directed generative models. However, upon personally examining the referenced study, I discovered that the particular quotation pertains specifically to one form of goal-directed generation method, namely Gaussian Process Bayesian Optimization (GP BO). In my opinion, the author has erroneously applied the quotation and has excessively emphasized the role of machine learning models within the domain of drug discovery.


**Documentation:**

I cannot open the URL to code the author provided.

**Limitations:**

See above

**Opportunities For Improvement:**

## Limited benchmarking effort
The study under review only incorporated two datasets related to affinity and overlooked the significance of ADMET (Absorption, Distribution, Metabolism, Excretion, and Toxicity) characteristics, which are of substantial importance in actual drug discovery processes. Also, the paper limited its focus to the testing of a handful of traditional machine learning models along with only two contemporary deep learning models. The existing benchmarking process appears to be neglecting a broad spectrum of advanced, efficient models currently available.

## Insufficient Rigor in Respect to Established Benchmarks
The manuscript at hand reports the presence of analogous molecules in certain tasks within existing benchmarks. Nonetheless, as I will elucidate in the section on correctness, this doesn't inherently imply that the previous benchmarks are inadequate for evaluating machine learning models in the context of drug discovery. Further, there are multiple tasks in most existing benchmarks, for example, 17 tasks in MoleculeNet. I don’t think two tasks containing similar molecules could impede their value. In my opinion, a minimum expectation should be to contrast model performance in prior benchmarks with the performance in the proposed benchmark. This comparison would serve to substantiate that the new benchmark is indeed more effective at distinguishing useful models.


**Relation To Prior Work:**

The relation to prior works is not sufficiently discussed, see above for details.

**Summary And Contributions:**

This manuscript proposes a benchmark named Lo-Hi, which is designed to test molecular property prediction models in drug discovery scenarios. The primary concept is to replicate the process of hit identification and lead optimization in drug discovery. This means focusing on out-of-distribution prediction in fit-identification and neighborhood prediction in lead optimization. The main contribution is a novel dataset splitting algorithm that resolves the Balanced Vertex Minimum k-Cut problem. However, I believe this paper lacks rigor in many areas. While it attempts to simulate the process of drug discovery, it still overlooks the complexities present in actual drug discovery, such as noise in the data, scarce and imbalanced nature of data, etc. It unduly diminishes other benchmarks while overhyping itself. Overall, I suggest that the authors focus on testing the dataset split method, conducting more targeted experiments, and resubmitting, instead of merely labeling it as a benchmark.

---

> ### Author Response · Authors · 2023-08-17
>
> Thank you for your comprehensive review. We appreciate your insights and have taken them into consideration. While we respectfully disagree on certain points, we have made revisions based on some of your suggestions.
>
> ### Overlooked complexities
>
> > it still overlooks the complexities present in actual drug discovery, such as noise in the data, scarce and imbalanced nature of data, etc.
>
> > A number of crucial facets of real drug discovery, such as data noise, data imbalance, data scarcity, and the multi-objective character of the drug, are overlooked in this benchmark.
>
> While no benchmark can capture all complexities of drug discovery, we disagree with the reviewer's criticisms:
>
> > noise data
>
> We respectfully disagree. Recognizing that datasets differ in noise levels, we specifically included more noisy IC50 datasets alongside less noisy Ki datasets. We also ensured the validity of the Lo dataset, constructing it so each cluster's variation exceeds experimental noise. Please refer to the main text (lines 274-278) and appendix B "Lo dataset is not just noise".
>
> > data imbalance
>
> We respectfully disagree. Real-life data varies in balance. We considered this by creating datasets with different balances (e.g. 0.67 positives in DRD2-Hi vs 0.04 in HIV-Hi). Refer to appendix A, table 3. We also took the extreme imbalance of HIV-Hi into account when training models, as noted in appendix H.8 "HIV-Hi balance.”
>
> > multi-objective drug discovery
>
> As mentioned in lines 13-16, a drug indeed must possess multiple properties simultaneously. We suggest only single-task datasets as it's common to train separate molecular property prediction models and apply them sequentially in virtual screening.
>
> > Data scarcity
>
> We concur. To enhance our paper, we introduced two datasets, KDR-Hi and KDR-Lo, both containing only 500 molecules in the training set to mimic low-data scenarios. We've also added to our future work section the idea of having smaller benchmarks, like FS-Mol, with fewer than 100 molecules but with a focus on Hi generalizability.
>
> ### Limited benchmarking effort
>
> > The study under review only incorporated two datasets related to affinity and overlooked the significance of ADMET (Absorption, Distribution, Metabolism, Excretion, and Toxicity) characteristics, which are of substantial importance in actual drug discovery processes
>
> We utilized not just two, but three datasets. The third is KCNH2, an ADMET target of cardiotoxicity assays. Furthermore, we have enhanced our paper by incorporating additional datasets. While we previously mentioned KDR-Hi and KDR-Lo, we have also introduced the Sol-Hi dataset focused on solubility. Solubility is a crucial ADMET property. We thank the reviewer for this suggestion.
>
> > Also, the paper limited its focus to the testing of a handful of traditional machine learning models along with only two contemporary deep learning models.
>
> Note that we tested the models not to select the best existing one, but to establish a firm baseline that isn't easily surpassed. While running numerous models with default hyperparameters is straightforward, comparing them fairly is challenging. We recommend [1, 2] for examples of unexpected outcomes from fair comparisons. With this in mind, we chose to limit the number of models we tested, but we ensured a thorough evaluation of each.
>
> ### Insufficient Rigor in Respect to Established Benchmarks
>
> > In my opinion, a minimum expectation should be to contrast model performance in prior benchmarks with the performance in the proposed benchmark. This comparison would serve to substantiate that the new benchmark is indeed more effective at distinguishing useful models.
>
> We did this in Appendix F, titled "Hi-split predicts virtual screening hit rate better than scaffold split". We demonstrate that using the same data, our new split predicts a model’s performance more accurately on novel molecules than the traditional scaffold split does. It also appears to select models with better metrics, though the difference isn't big.
>
> Another testament to the usefulness of the Lo/Hi splits is that different models excel at different tasks. If you mix Lo/Hi scenarios with traditional scaffold splits, you might identify only one "best" model, which could be suboptimal for your specific task (See Table 2).

---

> ### Author Response · Authors · 2023-08-17
>
> > The manuscript at hand reports the presence of analogous molecules in certain tasks within existing benchmarks. Nonetheless, as I will elucidate in the section on correctness, this doesn't inherently imply that the previous benchmarks are inadequate for evaluating machine learning models in the context of drug discovery. Further, there are multiple tasks in most existing benchmarks, for example, 17 tasks in MoleculeNet. I don’t think two tasks containing similar molecules could impede their value.
>
> As mentioned in lines 111-114, it's impossible to test every benchmark. That is why we opted to assess a variety of benchmarks using different data, different preprocessing techniques, and originating from different authors.
>
> To demonstrate that we aren't cherry-picking, we tested other MoleculeNet datasets. We've added this table to Appendix D, titled "Additional benchmarks analysis". Thank you.
>
> | Dataset | Fraction of Test Molecules Similar to Train Set |
> | --- | --- |
> | QM7 | 0.93 |
> | QM8 | 0.98 |
> | QM9 | 0.99 |
> | FreeSolv | 0.8 |
> | Lipophilicity | 0.67 |
> | PCBA | >0.93 |
> | MUV | 0.96 |
> | BACE | 0.77 |
> | Tox21 | 0.52 |
> | SIDER | 0.48 |
> | PDBBind (from DeepChem)* | 0.76 |
> | PDBBind (from MolNet)* | 0.83 |
>
> The table illustrates that the high similarity of molecules between training and testing is not exclusive to ESOL and HIV, which were studied in the main text.
>
> There are also a few more datasets in MolNet, however, their loading is currently not working in the DeepChem library (see the issues in the DeepChem repository). Hence, we opted not to analyze them in order to expedite our response.
>
> ### ****About the drug discovery pipeline****
>
> I believe there's been a misunderstanding. We aren't suggesting the exclusion of patented molecules from the training set. What we're emphasizing is that in Hit Identification, the ultimate goal is to discover a molecule with the desired properties that is distinct from those in the training set. It's a common practice to conduct virtual screening on molecules, filtering them based on their Tanimoto similarity to the training set [3-8]. The Hi split is designed to simulate this scenario, aiding in the selection of more suitable models.
>
> ### ****About goal-directed molecular generation****
>
> The reviewer notes that we may have misapplied the quotation from Gao et al.’s study, asserting that the quote pertained only to the GP BO model. Our interpretation of that quote and the corresponding paragraph differs. However, we believe the specifics aren't crucial, so we have removed the quotation. We hope this clarification enhances the paper's main message.
>
> ### Documentation
>
> We apologize for the broken link and have now uploaded a new version to OpenReview. This should remain accessible throughout the review process.
>
> While we maintain our stance on certain matters, your insights have undeniably enhanced our manuscript. Updates include:
>
> - Introduction of the ADMET-relevant Sol-Hi dataset.
> - Incorporation of the KDR-Hi and KDR-Lo datasets.
> - Comprehensive analysis of the majority of MolNet datasets.
> - Removal of the quotation from Gao et al.’s study for clarity.
>
> We appreciate the time and effort you've invested in reviewing our work and hope our revisions address your concerns.
>
> [1] Lucic, Mario, et al. "Are gans created equal? a large-scale study." *Advances in neural information processing systems* 31 (2018).
>
> [2] Musgrave, Kevin, Serge Belongie, and Ser-Nam Lim. "A metric learning reality check." *Computer Vision–ECCV 2020: 16th European Conference, Glasgow, UK, August 23–28, 2020, Proceedings, Part XXV 16*. Springer International Publishing, 2020.
>
> [3] Bender, Brian J., et al. "A practical guide to large-scale docking." *Nature protocols* 16.10 (2021): 4799-4832.
>
> [4] Stein, Reed M., et al. "Virtual discovery of melatonin receptor ligands to modulate circadian rhythms." *Nature* 579.7800 (2020): 609-614.
>
> [5] Stokes, Jonathan M., et al. "A deep learning approach to antibiotic discovery." *Cell* 180.4 (2020): 688-702.
>
> [6] Lyu, Jiankun, et al. "Ultra-large library docking for discovering new chemotypes." *Nature* 566.7743 (2019): 224-229.
>
> [7] Wong, Felix, et al. "Discovering small-molecule senolytics with deep neural networks." *Nature Aging* (2023): 1-17.
>
> [8] Liu, Gary, et al. "Deep learning-guided discovery of an antibiotic targeting Acinetobacter baumannii." *Nature Chemical Biology* (2023): 1-9.

---

### Author Response · Authors · 2023-08-17
**Manuscript revision**

We would like to express our gratitude to the reviewers for their invaluable feedback and suggestions. We have enhanced our paper based on these recommendations:

1. We incorporated two small additional datasets: KDR-Hi and KDR-Lo. Each of these datasets comprises 500 molecules in their training sets. They draw from activity data related to a kinase target, thereby enriching the diversity of our benchmark.
2. In response to suggestions to diversify our ADMET tasks, we introduced the solubility dataset Sol-Hi.
3. We conducted the hyperparameter tuning procedure once more for these datasets. Although GB/Chemprop remains the leading model for Sol-Hi, the KDR datasets present a different scenario. For KDR-Hi, Chemprop retains its top position, whereas for KDR-Lo, both Chemprop and SVM emerge as the frontrunners. However, it's worth noting that:
a) GB doesn't perform as well on KDR-Hi
b) Chemprop surpasses SVM on KDR-Lo
c) Both datasets exhibit unremarkable metrics. We believe the KDR datasets represent intriguing and challenging tasks.
4. We have incorporated an expanded analysis of the MoleculeNet datasets in Appendix D.
5. We have broadened our sections on limitations and potential future work.
6. We've made various minor modifications, including enhanced captions, grammar corrections, and the removal of a certain quote.

We hope these revisions address the concerns raised, and we appreciate the continued opportunity to improve our work.

---

### Decision · Program_Chairs · 2023-09-22

**Decision:**

Accept (Poster)

**Comment:**

The authors present two benchmarks for drug discovery modeled after two separate tasks in real-world drug discovery. To create the datasets, they propose a new variant of a graph cut algorithm. The authors give a critique of existing benchmarks and give practical results for ML models on their benchmark.

The reviewers at first glance had a high variance (3, 4, 7, 9). Two reviewers were very positive on the work. Both igzy and ZG59 mentioned the strong motivation for the benchmark, the clear writing, and informative case studies and experiments. igzy said “potentially very high impact and useful for the ML-biomedical boundary community, as it points out both major flaws and clear roadmaps for solutions in the important area of drug discovery datasets in a clearly written way, demonstrating substantial ML and domain knowledge.”

The two negative reviewers were less active in the discussion. Reviewer jiMQ made factual errors in their review and did not reply to the authors’ response. One of reviewer H78M’s main concerns was the lack of a feasibility/synthesizability task, however two other reviewers noted that this request is infeasible and beyond the scope of the paper.

After reading through the discussions, I agree with reviewers igzy and ZG59 about the high quality of this work, and the high potential for impact. I recommend acceptance.